# ATR kinase supports normal proliferation in the early S phase by preventing replication resource exhaustion

Demis Menolfi [1], Brian J. Lee [1], Hanwen Zhang[1], Wenxia Jiang [1], Nicole E. Bowen [2], Yunyue Wang[1], Junfei Zhao[3], Antony Holmes [1], Steven Gershik[1], Raul Rabadan [3], Baek Kim [2] & Shan Zha [1,4,5,6] ✉

The ATR kinase, which coordinates cellular responses to DNA replication stress, is also essential for the proliferation of normal unstressed cells. Although its role in the replication stress response is well defined, the mechanisms by which ATR supports normal cell proliferation remain elusive. Here, we show that ATR is dispensable for the viability of G0-arrested naïve B cells. However, upon cytokine-induced proliferation, Atr-deficient B cells initiate DNA replication efficiently, but by mid-S phase they display dNTP depletion, fork stalling, and replication failure. Nonetheless, productive DNA replication and dNTP levels can be restored in Atr-deficient cells by suppressing origin firing, such as partial inhibition of CDC7 and CDK1 kinase activities. Together, these findings indicate that ATR supports the proliferation of normal unstressed cells by tempering the pace of origin firing during the early S phase to avoid exhaustion of dNTPs and importantly also other replication factors.

Ataxia telangiectasia and Rad3-related (ATR) kinase is a member of the PI3K-related protein kinase (PI3KK) family that also includes Ataxia telangiectasia mutated (ATM) and DNA-dependent protein kinase (DNA-PK)[1]. Together, these three PI3KKs orchestrate the mammalian DNA damage response (DDR) by phosphorylating a wide variety of substrates that mediate appropriate cellular outcomes, such as DNA repair, cell cycle arrest, and/or cell death. While ATM and DNA-PK are enzymatically activated by DNA double-strand breaks (DSBs), ATR is induced primarily by replication stresses that generate single-strand DNA (ssDNA) structures. Since tumor cells often exhibit high levels of replication stress[2], pharmacological inhibitors of ATR are currently under development as potential cancer therapeutics. Interestingly, of the three PI3KKs, only ATR is required for animal development and the

normal proliferation of unstressed cells, implying that ATR has essential functions apart from its key role in the DDR[3–5]. Yet, despite extensive progress towards elucidating its DDR functions, we still do not understand how ATR supports replication in the absence of genotoxic stress and whether it does so through mechanisms related to its DDR activities. This question has become more pressing with recent reports of normal tissue toxicity in clinical trials of ATR inhibitors, which likely reflects their impact on the essential ATR functions of non-transformed cells[6].

In yeast, Mec1, the ATR ortholog in *S. cerevisiae*, is also required for both normal cell proliferation and the DDR to replication stress. Interestingly, in unstressed proliferating cells, Mec1[ATR] is transiently activated during early S phase by replication stress that arises

[1]Institute for Cancer Genetics, Vagelos College for Physicians and Surgeons, Columbia University, New York City, NY 10032, USA. [2]Department of Pediatrics, Emory University School of Medicine, Atlanta, GA 30322, USA. [3]Program for Mathematical Genomics, Department of Systems Biology, Vagelos College for Physicians and Surgeons, Columbia University, New York City, NY 10032, USA. [4]Department of Pathology and Cell Biology, Herbert Irvine Comprehensive Cancer Center, Vagelos College for Physicians and Surgeons, Columbia University, New York City, NY 10032, USA. [5]Division of Pediatric Hematology, Oncology and Stem Cell Transplantation, Department of Pediatrics, Vagelos College for Physicians and Surgeons, Columbia University, New York City, NY 10032, USA. [6]Department of Immunology and Microbiology, Vagelos College for Physicians and Surgeons, Columbia University, New York City, NY 10032, USA. ✉e-mail: sz2296@cumc.columbia.edu

spontaneously when deoxyribonucleotide triphosphate (dNTP) levels become insufficient to support DNA replication[7]. Activated Mec1[ATR] in turn induces a kinase cascade involving Rad53[CHK1] and Dun1 kinases that promote degradation of Sml1, a negative regulator of ribonucleotide reductase (RNR), allowing for de novo generation of dNTPs and continued DNA replication[8,9]. Indeed, the growth of Mec1-deficient yeast can be rescued by deleting Sml1, establishing Sml1 regulation as an essential Mec1[ATR] function for normal DNA replication[10]. Of note, ATR has also been implicated in transcriptional upregulation of RNR in response to replication stress[11]. However, functional orthologs of Sml1 and Dun1 have not been found in mammals, leaving the essential role of ATR in the normal replication of mammalian cells still unresolved.

Upon replication stress, the ATR kinase and its obligatory partner, ATR-interacting protein (ATRIP), are recruited to and activated by RPA-coated ssDNA filaments generated by DSB end resection or by various processes that interfere with DNA replication (e.g., R-loop formation, fragile site expression)[11-16]. Full activation of the ATR kinase also requires its association with either the TOPBP1 or ETAA1 proteins[17-20]. Stress-activated ATR phosphorylates many substrates, including its effector kinase CHK1, which in turn elicit the appropriate cellular responses, including suppression of firing of dormant replication origins[21], induction of cell cycle arrests at the S/G2[22] and G2/M transitions[23,24], stabilization of stalled replication forks[25,26], and the eventual restart of DNA replication[3]. However, we do not know whether any of these stress-induced pathways also mediate the essential functions of ATR in the proliferation of unstressed cells.

Investigation of ATR essential functions has been technically challenging, especially given the long half-life (>24 h) of ATR protein[27]. Nonetheless, somatic cells that reside in the G0 phase of the cell cycle for extended periods, such as peripheral lymphocytes, should provide a suitable genetic platform to examine these aspects of ATR biology. Here we use the development and activation of lymphocytes as model systems to study the essential physiological role of ATR in normal replication. Developing B lymphocytes undergo several rounds of proliferation after successfully assembling the Ig heavy chain and before assembling their Ig light chain gene, after which lymphocytes migrate to secondary lymphoid organs (e.g., spleen and lymph nodes), where they remain quiescent for months or years. Unlike tissue culture cells, which undergo continuous growth without an obvious G0 phase, peripheral lymphocytes reside in G0. Upon antigen exposure and cytokine stimulation, lymphocytes take ~24 h to enter the first round of DNA replication and then replicate 4–8 times over the next 2–3 days[28], providing synchronized and well-defined G0-G1-S phase transitions in which to investigate ATR function in the absence of external replication stress. Using developmental stage specific Cre and conditional ATR alleles, we found that while ATR is dispensable for both the initiation and maintenance of DNA replication, it is required to prevent the exhaustion of replication factors, including nucleotides, during early S phase. Unlike the effects of Mec1[ATR] loss in yeast, nucleotide shortage in ATR-deficient lymphocytes is caused by overuse of cellular dNTP pools rather than limited dNTP production. In accord with these results, dNTP levels, cell cycle progression, and replication fork-related phenotypes of ATR-deficient B cells can be rescued by treatments that partially suppress new origin firing, such as inhibition of CDC7 and CDK1 kinase activities. Together, these data establish that ATR facilitates normal proliferation by balancing the pace of DNA replication with the availability of replication resources, providing the mechanism underlying the normal tissue toxicity of ATR kinase inhibitors.

## Results

### ATR is essential for early B cell development and dispensable in naïve B cells

To analyze the role of ATR in normal primary cells, we inactivated the *Atr* gene using either *Mb1^Cre^* (*CD79a*), which expresses Cre-recombinase in pre-B cells before clonal expansion and light chain

gene rearrangement[29], or *CD21-Cre*, which expresses Cre-recombinase in naïve B cells after successful light chain gene rearrangement and surface IgM expression[30] (Supplementary Fig. 1a). Cre-positive *Atr^+/C^* mice were included as controls to ensure that the observed phenotypes are truly caused by Atr inactivation rather than Cre-recombinase expression or DNA damage caused by Cre-mediated excision. While control *Mb1^+/Cre^ Atr^+/C^* mice undergo normal B cell development, the percentage of naïve B cells (B220+IgM + ) were dramatically reduced in the bone marrow and spleen of *Mb1^+/Cre^Atr^C/-^* and *Mb1^+/Cre^Atr^C/KD^* mice. Given the restricted deletion of Atr in B cells, but not T cells or other hematopoietic cells in *Mb1^+/Cre^* mice, these results indicate an essential role for ATR in B cell development, likely the proliferation and clonal expansion before Ig light chain rearrangements (Fig. 1a, b)[4]. In contrast, when Atr DNA and protein loss are restricted to naïve B cells after successful Ig light chain assembly, as in *CD21-Cre^+^Atr^C/-^* and *CD21-Cre^+^Atr^C/KD^* mice (Supplementary Fig. 1b), B220 + B cells frequencies and counts remain normal at all subsequent developmental stages, indicating that ATR protein and its kinase activity are dispensable for the viability of quiescent naïve B cells (Fig. 1a–c)[4].

Upon antigen exposure, naïve B cells exit G0, rapidly proliferate, and undergo IgH class switch recombination (CSR) to generate antibodies with different effector functions[31]. This process can be modeled in purified splenic B cells with a cocktail of cytokines, providing a system to study the essential role of ATR in normal proliferation of primary cells. Specifically, IL-4 and anti-CD40 activate naïve B cells and initiate robust CSR to IgG1 (~30% IgG1+ at day 4) and IgE. However, the Atr-deficient B cells from *CD21-Cre^+^Atr^C/-^* and *CD21-Cre^+^Atr^C/KD^* mice display a ~ 50% decrease in CSR to IgG1 (Fig. 1d, e) and a massive loss of viability due to concurrent apoptosis indicated by positive Annexin-V or PI staining, after day 2 (Supplementary Fig. 1c–f). Reduced CSR can be caused by DNA end-ligation defects, faulty proliferation, or both. To determine whether ATR has a role in the end-ligation phase of CSR, we analyzed CSR junctions using high-throughput genome-wide translocation sequencing (HTGTS)[32]. While junctions recovered from non-homologous end-joining (NHEJ)-deficient B cells (*e.g., DNA-PKcs^-/-^, Xrcc4^-/-^*) displayed extensive resection and preferential usage of microhomology[33-35], junctions from *CD21-Cre^+^Atr^C/-^* and *CD21-Cre^+^Atr^C/KD^* B cells are indistinguishable from those recovered from control *CD21-Cre^+^Atr^+/C^* B cells (Supplementary Fig. 1g–i)[36]. Nonetheless, cell proliferation of activated *CD21-Cre^+^Atr^C/-^* and *CD21-Cre^+^Atr^C/KD^* B cells was severely impaired (Fig. 1d), suggesting that *Atr* supports B cell CSR by ensuring efficient proliferation. To determine the physiological impact of Atr in B cell activation, we examined germinal center formation, where B cells expand and mature after antigen exposure with the help of T cells[37]. Thus, the germinal center B cells (CD95+ and GL7 + B cells) represent the most actively proliferating B cells. In comparison to age-matched *CD21-Cre^+^Atr^+/C^* control mice, CD*21-Cre^+^Atr^C/-^* mice have reduced germinal center B cells frequency (among all splenic B cells) and fewer germinal centers (stained positive for Bcl6, a germinal center marker) both before and after immunizations (Fig. 1f, g), indicating compromised B cell clonal expansion in vivo. To understand whether this role of ATR in primary B cells in G0 is also conserved in transformed B cells, we generated v-abl kinase transformed B cell lines carrying the tamoxifen inducible *Rosa^ER-CreT2^Atr^+/C^* or *Rosa^ER-CreT2^Atr^C/KO^* alleles. Addition of Tamoxifen rapidly activated *Atr* deletion and caused a delayed and more moderate reduction in viability in *Rosa^ER-CreT2^Atr^C/KO^* transformed B cells (Supplementary Fig. 1j, k), highlighting a consistent and potentially more prominent role of ATR in initiating proliferation in G0 arrested primary cells.

### ATR kinase activity is essential for productive DNA replication

To understand how ATR supports the proliferation of primary B cells, we analyzed cell cycle progression at 48 h after cytokine stimulation, a time when viability has not yet been severely affected by Atr deficiency (Supplementary Fig. 1c–f). After pulse labeling with BrdU (30 min), <5%

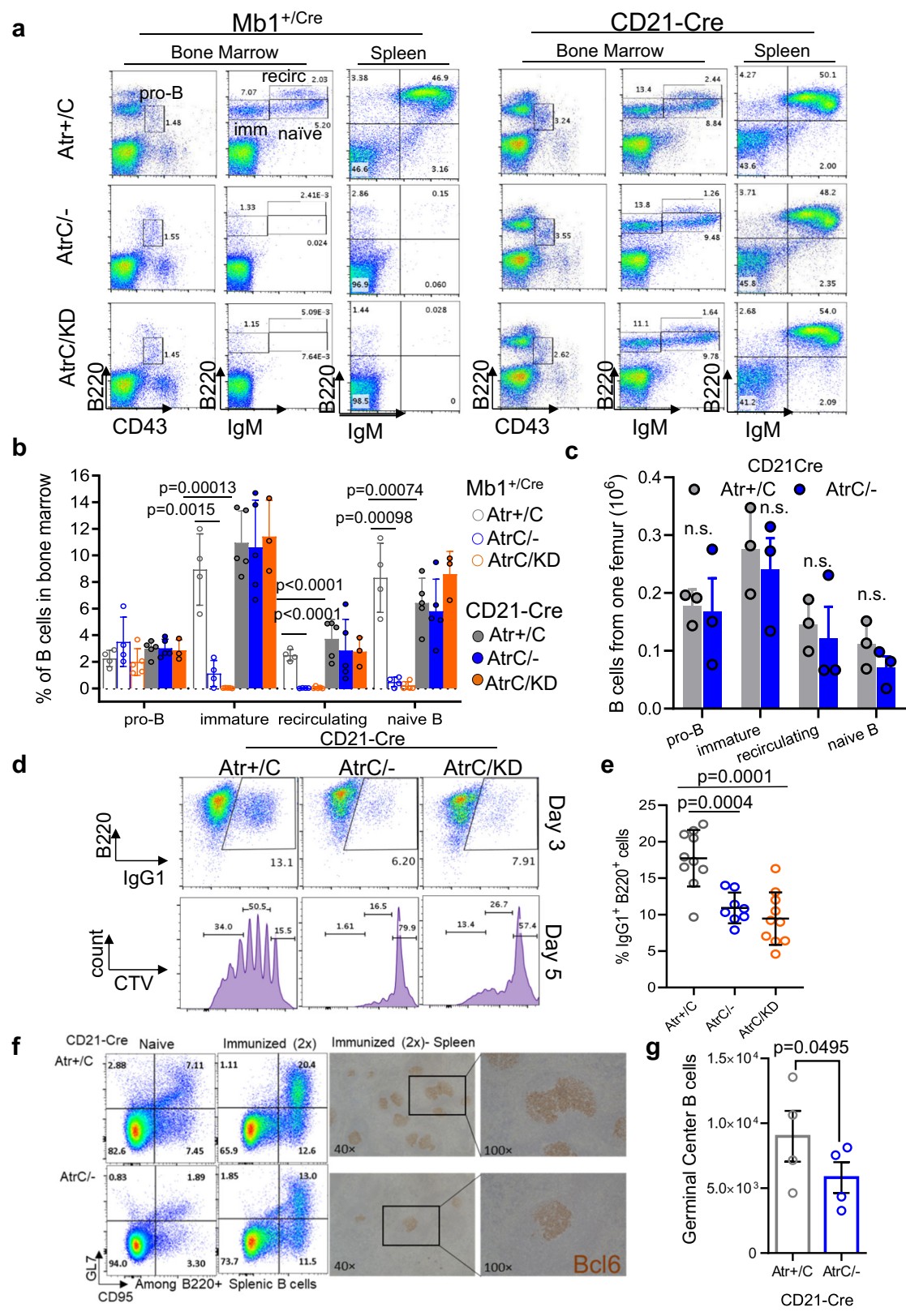

of the mid-S phase (as determined by PI staining) control *CD21-Cre⁺Atr^{+/C}* B cells were BrdU-negative (BrdU-). In contrast, roughly half of the mid-S phase *CD21-Cre⁺Atr^{C/-}* and *CD21-Cre⁺Atr^{C/KD}* cells were BrdU- (Fig. 2a) and remained so at 72 and 96 h post-stimulation (Supplementary Fig. 2a, b). A similar phenotype, but to a lesser degree, was seen in transformed B cells at 4 days after Tamoxifen induced deletion of ATR (Supplementary Fig. 1j). We reasoned that the B cell line in the

continued culture might have more reserve of nucleotides and replication factors than primary B cells, which enter the cell cycle from G0. This striking loss of BrdU incorporation into the genome is consistent with massive fork stalling. Indeed, DNA fiber analyses revealed a severe reduction of IdU track length in replicating *CD21-Cre⁺Atr^{C/-}* and *CD21-Cre⁺Atr^{C/KD}* cells (Fig. 2b and Supplementary Fig. 2c). Moreover, *CD21-Cre⁺Atr^{C/-}* cells displayed a significant increase in asymmetric forks

**Fig. 1 | ATR is essential for early B cell development and dispensable in naïve B cells. a** Bone marrow and spleens from $Mb1^{+/Cre}$ $Atr^{+/C}$, $Atr^{C/-}$ and $Atr^{C/KD}$ and $CD21\text{-}Cre^+$ $Atr^{+/C}$, $Atr^{C/-}$ and $Atr^{C/KD}$ mice were harvested, and cells stained with the antibodies indicated. Pro-B, immature, recirculating, and naïve B cells were gated in the bone marrow, as shown in the examples. **b** Quantification of B cell percentage in bone marrow from three, four or five biologically independent mice for both the $Mb1^{+/Cre}$ and $CD21\text{-}Cre^+$ alleles. Data are presented as mean values ±SEM. Two-tailed t test was used. **c** Quantification of absolute B cell number from one femur from three biologically independent $CD21\text{-}Cre^+$ Atr proficient or deficient mice. Mean and standard errors were plotted. **d** $CD21\text{-}Cre^+$ $Atr^{+/C}$, $Atr^{C/-}$ and $Atr^{C/KD}$ B cells were purified from the mice spleens and cultured with IL-4 and anti-CD40 for CSR. Activated B cells were stained on Day 3 for IgG1 and B220 markers. Flow cytometry profiles are shown as examples. Cells were stained with Cell Trace Violet (CTV) at the beginning of the CSR experiment and analyzed for proliferation on Day 5 post-stimulation. Quantification divides CTV-stained cells into three populations, from right to left: 0–1, 2–4, and ≥5 generations. **e** Quantification of CSR experiments from ten $CD21\text{-}Cre^+$ $Atr^{+/C}$, eight $CD21\text{-}Cre^+$ $Atr^{C/-}$ and ten $CD21\text{-}Cre^+$ $Atr^{C/KD}$ biologically independent mice. Data are presented as mean values ±SD. **f** Flow cytometry analyses of spleen germinal center B cells (CD95+ and GL7+) in naïve and immunized (with sheep red blood cells twice) $CD21\text{-}Cre^+$ $Atr^{+/C}$ and $Atr^{C/-}$ mice. Representative immunohistochemistry staining for Bcl6, a germinal center marker, are included on the right. **g** Quantification of splenic germinal center B cell number after two immunizations from four biologically independent $CD21\text{-}Cre^+$ $Atr^{+/C}$ or $CD21\text{-}Cre^+$ $Atr^{C/-}$ mice. Mean and standard errors were plotted. Statistical analyses in **b**, **c**, **e**, and **g** were done using unpaired two-tailed Student's $t$ test. Source data are provided as a Source Data file.

(Supplementary Fig. 2d), consistent with the presence of replication blocks or barriers previously reported on CHK1 inhibitor-treated cells[38]. These cells also displayed extensive γH2AX signals measured by flow cytometry (Fig. 2c) and increased DNA breaks in alkaline comet assays (Fig. 2d), consistent with fork collapse, as well as a robust DNA damage response characterized by the hyper-phosphorylation of KAP1 and RPA (Fig. 2e). FISH analyses with a telomere probe that helps identify chromosomal breaks in short mouse chromosomes[39] revealed a moderate increase in breaks and fragile telomeres in $CD21\text{-}Cre^+Atr^{C/-}$ or $CD21\text{-}Cre^+Atr^{C/KD}$ B cells at D3 of CSR (Supplementary Fig. 2f). However, since the analyses were performed on metaphase preparations and there are pronounced S phase progression defects in $Atr\text{-}deficient$ B cells (Fig. 2a), we reasoned that the chromosomal breaks and telomere instability are markedly underestimated in $CD21\text{-}Cre^+Atr^{C/-}$ or $CD21\text{-}Cre^+Atr^{C/KD}$ B cells. Finally, unlike the control $CD21\text{-}Cre^+Atr^{+/C}$ B cells, $CD21\text{-}Cre^+Atr^{C/-}$ or $CD21\text{-}Cre^+Atr^{C/KD}$ B cells lacked phosphorylated CHK1, consistent with the absence of ATR kinase activity. Given that mouse cells have 50-fold lower DNA-PKcs protein levels than their human counterparts[40], this might explain why we did not see any DNA-PK-mediated compensatory phosphorylation of CHK1[11] as reported in human cancer cells treated with ATR inhibitor. Overall, the phenotypes of $CD21\text{-}Cre^+Atr^{C/-}$ and $CD21\text{-}Cre^+Atr^{C/KD}$ cells are indistinguishable, indicating that the kinase activity of ATR is required for proliferation.

## ATR kinase activity is required at the early S phase to support productive replication

To pinpoint at which stage of B cell proliferation ATR kinase activity is required, we treated Atr-proficient B cells with the ATR kinase inhibitor (ATRi) VE-821 at various time points after cytokine stimulation and then measured BrdU incorporation at mid-S phase (48 h post-stimulation). Although ATRi treatment (5 μM) at the beginning of stimulation (time zero) abrogated BrdU incorporation, ATRi treatment at 46 h post-stimulation (for only 2 h before and during the BrdU labeling), even at higher concentrations (10 μM), had no effect on BrdU incorporation (Fig. 2f). This data suggest that on going S phase cells do not require ATR kinase activity for productive replication. Instead, ATR is required for the normal proliferation of unstressed cells before the mid-S phase, either at the G1/S transition or during the early S phase. This is in contrast to the well-characterized role of ATR in the replication stress response, in which ATR activity is needed after DNA damage (e.g., HU or cisplatin)[14]. Moreover, this BrdU-negative phenotype is elicited specifically by ATR inhibition (Fig. 2f), as it is not observed upon inhibition of ATM (KU55933) or DNA-PKcs (NU7441) with well-characterized commercial inhibitors or by exposure to low-dose hydroxyurea (HU, 20 μM), which is known to induce robust ATR activation but still allows S phase entry (Supplementary Fig. 2g). High dose of HU (2 mM) completely abolished B cell replication and S phases as seen by others[41] (Supplementary Fig. 2h). Consistent with these findings, B cells lacking ATM[42–44] or DNA-PKcs[34,45,46] do not show significant proliferation defects. Finally, the v-abl kinase transformed $Rosa^{ER\text{-}CreT2}Atr^{C/KO}$ B cells also accumulated BrdU-negative S phase cells,

suggesting this phenotype is consistent in transformed cells (Supplementary Fig. 1j).

To rule out DNA damage generated from the first round of replication in the absence of ATR as a cause for the BrdU-negative phenotype at 48 h after stimulation, we performed the experiments at 24 h when B cells enter their first S phase (Supplementary Fig. 3a). Focusing on the first 24 h also avoided the expression of Activation-induced deaminase (AID), which can potentially introduce additional genomic instability[47,48]. The percentage of total BrdU-positive cells was similar for Atr-deficient as for the Atr-proficient B cells (Fig. 3a, b), suggesting that Atr is dispensable for replication initiation (Supplementary Figure 3a, b). But, while ~60% of BrdU+ control cells have mid and late S phase and G2 DNA content (measured by PI staining), >70% of BrdU+ Atr-deficient lymphocytes have only G1 DNA content (Fig. 3a, c), indicating the lack of productive DNA replication. Like at 48 h, nearly half of 24 h S phase Atr-deficient lymphocytes are BrdU negative (Fig. 3d). Moreover, cells treated with ATRi before S phase onset, $t = 0$ h, and $t = 10$ h time points display a similar early S phase arrest as $CD21\text{-}Cre^+Atr^{C/-}$ B cells. In contrast, cells treated with ATRi in the middle of the first S phase (22 h after activation, 2 h before BrdU labeling) do not show any replication block or significant BrdU-neg fraction (Supplementary Fig. 3c), suggesting the ongoing fork does not need ATR kinase activity for immediate support. Pulse-chase labeling with BrdU at 24 h showed that BrdU-positive $CD21\text{-}Cre^+Atr^{+/C}$ cells proceeded through G2 and the next G1. In contrast, BrdU+ $CD21\text{-}Cre^+Atr^{C/-}$ B cells are stalled and cannot complete replication and do not return to G1 for at least 24 h, a phenotype that was reproduced by the ATRi supplemented at the time of activation (BrdU pulse-labeling after 24 h of ATRi and then chase another 24 h with ATRi, total ATRi 0–48 h, blue arrow). However, when ATRi was added later in S phase (BrdU pulse-labeling after transient exposure to ATRi for 30 min at 24 h after activation, than chase for additional 24 h during 24–48 h after activation with ATRi, total ATRi from 24–48 h after activation, red arrow) or for 1 hour and then washed away (chase for 24 h without ATRi after pulse lable for 30 min with ATR at 24 h after activation, green arrow) cells proceeded through the cell cycle as control cells (Fig. 3e and quantification at Supplementary Fig. 3d). This finding also suggests that ATR kinase is dispensable for ongoing S phase, and seems more critical before productive DNA replication starts – such as for G0 cells entering their first replication. Furthermore, early S phase $CD21\text{-}Cre^+Atr^{C/-}$ and $CD21\text{-}Cre^+Atr^{C/KD}$ cells are positive for the mitotic marker Histone H3 Ser10 phosphorylation, suggesting premature CDK1 activation (Fig. 3f and Supplementary Fig. 3e). To understand the cause of the non-productive DNA replication in ATR-deficient cells at the early S phase, we defined the time of cytokine addition as G0 since purified B cells are quiescent. The 10 h time point was chosen to represent G1 cells without significant BrdU incorporation (Fig. 3g). At the G1 phase, the cells have initiated amino acid synthesis and ribosomal biogenesis to prepare for proliferation (see below, Supplementary Figs. 4a and 5a, b). Curiously, significantly more DNA strand breaks measured by alkaline comet assay were detected in the quiescent stage

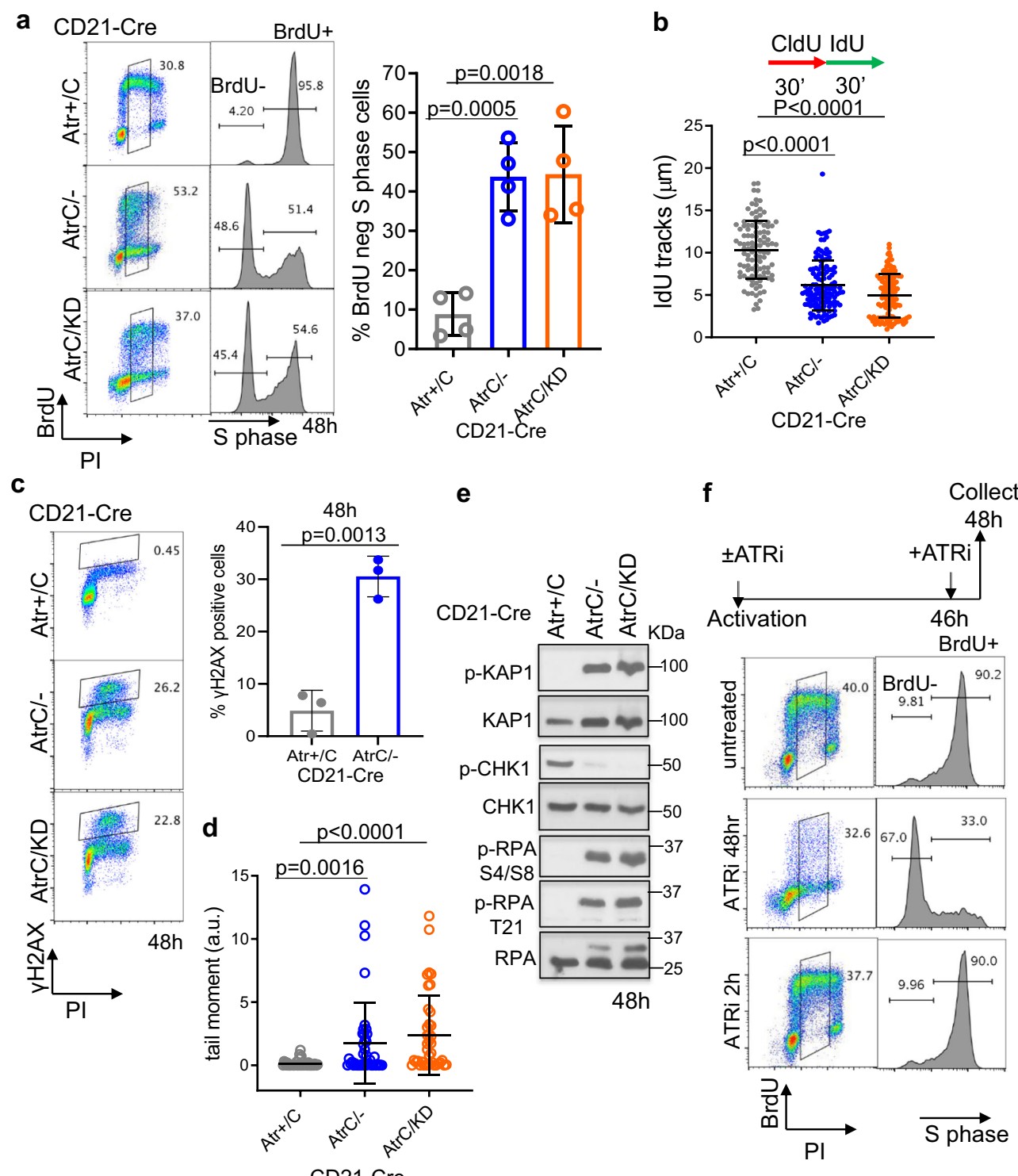

(denoted as G0, 0 h) and in G1 (10 h) *CD21-Cre*⁺*Atr*^C/- and *Atr*^C/KD cells in comparison to the *CD21-Cre*⁺*Atr*^+/C control (Fig. 3h. Neutral comet assay at 10 h post-stimulation showed that at least a subset of the breaks were DSBs (Supplementary Fig. 3f). In this regard, ATR has been implicated nucleoside excision repair and might be activated in some special cases by ssDNA generated during transcription[1,27]. Meanwhile, CHK1 kinase is undetectable in G0, very low in G1 (10 h), and accumulated and phosphorylated in the S phase at 24 h and 48 h (Supplementary Fig. 3g). Moreover, hyper-phosphorylation of KAP1 and RPA, two markers of ATM activation, are only apparent in Atr-deficient cells

after 24 h of activation (Supplementary Fig. 3h). These data suggest an important function of ATR in preserving genome integrity before or at the early S phase to support efficient DNA replication.

## Loss of ATR kinase does not affect the S phase transcriptional program

Since the replication phenotypes of the *CD21-Cre*⁺*Atr*^C/- and *Atr*^C/KD cells are indistinguishable, we focused on *CD21-Cre*⁺*Atr*^C/- cells. First, we performed RNA-seq analyses on samples collected at 0, 10, and 24 h after stimulation. Gene Ontology (GO) analysis revealed that the gene

**Fig. 2 | ATR kinase activity is essential for productive DNA replication. a** Flow cytometry profiles of *CD21-Cre⁺ Atr⁺/ᶜ, Atrᶜ/⁻,* and *Atrᶜ/ᴷᴰ* B cells pulse-labeled with BrdU for 30 min at 48 h post-stimulation. Representative dot plots are shown. Histogram plots of S-phase cells are shown with the separation in BrdU-negative (BrdU−) and BrdU-positive (Brdu+) cells. Quantification of the percentage of BrdU negative cells from four independent experiments is reported, together with the two-tailed t-test statistical analysis. Data are presented as mean values ±SEM. **b** *CD21-Cre⁺ Atr⁺/ᶜ, Atrᶜ/⁻,* and *Atrᶜ/ᴷᴰ* B cells were labeled with CldU for 30 min and with IdU for an additional 30 min on day 2 of CSR, and DNA fibers were spread and stained. The length of IdU fibers was measured using Image J and plotted as shown. Two-sided Mann-Whitney test was used. Data are presented as mean values ±SD. **c** *CD21-Cre⁺ Atr⁺/ᶜ, Atrᶜ/⁻,* and *Atrᶜ/ᴷᴰ* B cells at 48 h post-stimulation were stained for γH2AX and PI and analyzed by flow cytometry. Quantification of three independent experiments of γH2AX/PI staining is shown for *CD21-Cre⁺ Atr⁺/ᶜ* and *Atrᶜ/⁻* B cells, with two-tailed *t*-test analysis indicated. Data are presented as mean values ±SEM. **d** Alkaline comet assay was performed in *CD21-Cre⁺ Atr⁺/ᶜ Atrᶜ/⁻* and *Atrᶜ/ᴷᴰ* B cells at 48 h post-stimulation. One representative of two independent experiments is shown. The tail moment (arbitrary units, a.u.) of single cells is reported with two-taled t-test analysis indicated. Data are presented as mean values ±SD. **e** *CD21-Cre⁺ Atr⁺/ᶜ, Atrᶜ/⁻,* and *Atrᶜ/ᴷᴰ* B cells were lysed on day 2 of CSR, and the indicated proteins were analyzed by western blot. **f** 48 h stimulated WT B cells were left untreated, treated with 5 μM ATRi (VE-821) from the time of B cells activation or with 10 μM ATRi (VE-821) for 2 h right before BrdU labeling (30 min) and collection (immediately after labeling). Source data are provided as a Source Data file.

expression programs in ATR-proficient and -deficient cells are comparable. From G0 (0 h) to G1 (10 h), several RNA-related pathways are upregulated independent of Atr status (Supplementary Fig. 4a). From G1 (10 h) to S (24 h) phase, *CD21-Cre⁺Atr⁺/ᶜ* and *CD21-Cre⁺Atrᶜ/⁻* cells upregulate genes involved in DNA replication and chromosome segregation (Fig. 4a). Importantly, *Atr*-deficient B cells upregulate origin loading factors (*e.g.*, Orc2−) in G1 and replication factors (e.g., Mcm, Cdc, replicative polymerases) in the S phase at levels comparable to control *CD21-Cre⁺Atr⁺/ᶜ* cells (Fig. 4b). Gene Set Enrichment Analysis (GSEA) also further confirmed that *CD21-Cre⁺Atrᶜ/⁻* cells efficiently initiate a transcriptional program associated with DNA replication, indicating that ATR is dispensable for initiating the initiation of DNA replication (Supplementary Fig. 4b).

## Metabolome profiling and direct dNTP measurement reveal selective nucleoside and deoxyribonucleotide defects in Atr-deficient B cells

In yeast, ATR ortholog Mec1 plays a critical role in regulating deoxyribonucleotide syntheses through post-translation modification of RNR[8,9,49]. To understand whether mammalian ATR also regulates nucleotide metabolism to support DNA replication, we profiled polar metabolites in G0 (0 h), G1 (10 h), and S (24 h) phase B cells via mass spectrometry[50] (Fig. 5a). Principal component analysis (PCA) shows that G0, G1, or S phase B cells can be clearly distinguished in all the samples analyzed (Supplementary Fig. 5a). Pathway impact analyses showed that the accumulation of non-essential amino acids, the tricarboxylic acid (TCA) cycle metabolites, glycolysis, and gluconeogenesis are the most noticeable changes in G1 from G0 phase cells (Fig. 5b). At the same time, purine and pyrimidine metabolism are among the most incremented pathways between G0 and S (Fig. 5c). As previously documented for T cells[51], activated B cells first generate amino acids and then nucleotides. The increases of TCA cycle metabolites and amino acids from the G0 to G1 phase are not affected by ATR status (Supplementary Fig. 5b). In contrast, multiple nucleotide precursors and intermediates (e.g., ribose-phosphate, monophosphate nucleosides -IMP, AMP, GMP, UMP, CMP, diphosphate nucleosides − IDP, ADP, GDP, UDP, CDP, and their deoxy forms) are significantly lower in 24 h (S phase) ATR-deficient cells than in control ones (Fig. 5d), suggesting a selective deficit in nucleotides. However, the abundance of the triphosphate nucleotides (e.g., ATP, GTP, and CTP) is slightly higher in Atr-deficient cells, potentially reflecting a usage defect or an excessive conversion of the NMP and NDP to the NTP forms. Of note, since ribonucleotide levels are hundreds of folds higher than deoxyribonucleotides, the bulk metabolome predominantly reflects ribonucleotide changes. To understand whether the levels of dNTPs required for DNA replication are compromised by ATR deficiency, we measured dNTPs in vivo using a previously described enzymatic method[52] (Fig. 5a). As expected, dNTP levels rise sharply in the S phase (24 h) control cells (Supplementary Fig. 5c). Atr-deficiency markedly decreases all four dNTP levels in G1 as well as S phase cells (Fig. 5e, f). In addition, control cells treated with ATRi also

displayed a dramatic dNTP deficit at levels comparable to *CD21-Cre⁺ Atrᶜ/⁻* cells (Fig. 5f). Moreover, dNTP levels in quiescent cells were very low compared to G1 and S phases and similar between control and ATR-deficient cells (Supplementary Fig. 5d). Finally, RNA-seq analyses confirmed a robust and comparable activation of nucleotide metabolism gene expression in *ATR-deficient* cells (Supplementary Fig. 5e). Consistent with a recent report in T cells treated with ATR inhibitors[53], we noted a moderate but not statistically significant decrease in the induction of RNR subunits (Rrm1 and Rrm2) in ATR-deficient B cells in S phase (Supplementary Fig. 5e, from -1.5 folds to -1.3 folds). The significance of this reduction will be tested and discussed later.

## Deoxyribonucleosides do not rescue replication fork stalling but alleviate the DNA damage response in ATR-deficient cells

Lymphocytes express high levels of deoxycytidine kinase (dCK) that can convert deoxynucleoside (dN) to deoxynucleotide (dNTP) therefore bypassing RNR. Deletion of dCK specifically blocks B and T cell development without overt phenotypes in other organs[54]. Taking advantage of this, we tried to rescue ATR-deficient phenotypes in B cells by supplementing cells with deoxyribonucleosides – deoxyguanosine, deoxyadenosine, deoxycytidine, and thymidine. Direct measurements of intracellular levels of dNTPs confirmed the successful uptake and phosphorylation by the cells (Fig. 6a and Supplementary Fig. 6a), making the levels of dNTPs now comparable between *CD21-Cre⁺Atr⁺/ᶜ* and *Atrᶜ/⁻* B cells. Nevertheless, they failed to rescue viability and CSR defects of ATR-deficient cells (Supplementary Fig. 6b). DNA replication progression measured by BrdU incorporation in *CD21-Cre⁺Atrᶜ/⁻* B cells was also not improved with deoxyribonucleosides alone or when supplemented with additional ribonucleosides (Fig. 6b and Supplementary Fig. 6c). We noted that the balanced nucleoside and deoxynucleoside supplement does not affect S phase progression in normal cells[55], suggesting the failure in rescuing B cell replication cannot be explained by nucleotide imbalance alone[53]. Together, these data suggest that ATR regulates productive DNA replication at levels beyond dNTPs and NTPs in B lymphocytes. In this context, previous studies have also tried and failed to use nucleoside (ribose form) to rescue the proliferation of CHK1-inhibitor-treated human cancer cells[38]. And other studies have suggested that upon HU-induced replication stresses, ATR inhibitor causes RPA exhaustion[14], proposing an additional rate-limiting factor in ATR-deficient cells.

Interestingly, the supplementation of deoxyribonucleosides at 10 h post-stimulation, despite not being able to restore the replication forks, alleviated the DNA damage response in ATR-deficient cells. Indeed, hyper-phosphorylation of KAP1, RPA, and γH2AX were all extensively reduced in deoxyribonucleoside supplemented *CD21-Cre⁺Atrᶜ/⁻* cells (Fig. 6c), as well as the accumulation of DNA damage detected by comet assay (Fig. 6d). The degree of rescue positively correlated with the amount of deoxyribonucleosides provided to the culture, supporting a model in which nucleotides are one of the many

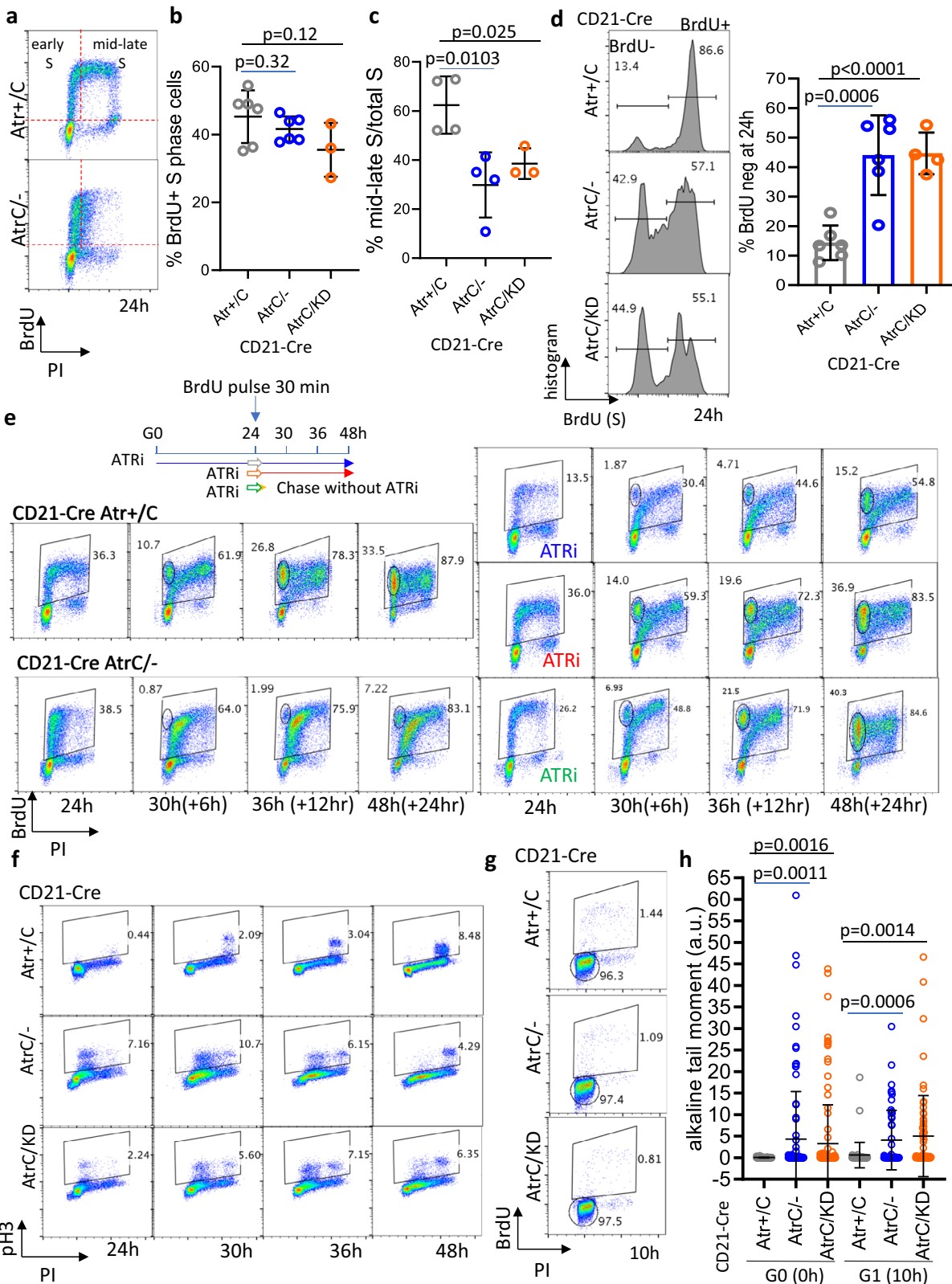

rate-limiting factors for productive DNA replication in ATR-deficient cells. As such, nucleotide supplementation seems necessary but insufficient to fully restore productive DNA replication.

**CRISPR-Cas9 Screen in ATRi-treated murine transformed B cells**

To determine the mechanisms by which ATR mediates its essential function in normal cell proliferation, we reasoned to investigate the

pathways that allow cells to survive ATR inhibition. Therefore, we conducted unbiased CRISPR-Cas9 screens of a lentiviral gRNA library (18,424 mouse genes) in murine B cells exposed to high doses (IC90) of ATR inhibitors (VE-821 for two independent screens, and AZD6738). In addition, since the ATR substrate CHK1 is likewise essential for normal cell proliferation[23,56], we also conducted screens using a CHK1 inhibitor (LY2603618) (Fig. 6e). In contrast to previous screens performed

**Fig. 3 | ATR is required at G1-S transition to support productive replication.** **a** *CD21-Cre⁺ Atr⁺/C* and *Atr^C/-* B cells were pulse-labeled with BrdU for 30 min at 24 h post-stimulation. Representative flow cytometry profiles are shown. **b**) Quantification of the percentage of BrdU+ S phase cells collected from six *CD21-Cre⁺ Atr⁺/C*, six *CD21-Cre⁺ Atr^C/-* and three *CD21-Cre⁺ Atr^C/KD* biologically independent mice. Data are presented as mean values ±SD. Two-tailed *t*-test. **c** Quantification of the percentage of BrdU+ mid-late S phase cells relative to total S phase cells from cell cycle experiments in B cells collected from six *CD21-Cre⁺ Atr⁺/C* mice, six *CD21-Cre⁺ Atr^C/-*, and three *CD21-Cre⁺ Atr^C/KD* biologically independent mice. Data are presented as mean values ±SD. Two-tailed *t*-test. **d** Six *CD21-Cre⁺ Atr⁺/C*, six *Atr^C/-*, and four *Atr^C/KD* biologically independent B cell samples, treated as in **a**, were analyzed at 24 h post-stimulation for the percentage of BrdU negative and positive S phase cells. Data are presented as mean values ±SEM. Two-tailed *t*-test. **e** Schematic representation of the BrdU pulse-chase experiments. Cells were pulse labled with BrdU for 30 min at

24 h after stimulation and chased for another 24 h. *CD21-Cre⁺ Atr^C/C* cells were treated with 5 μM ATRi at time 0 (blue, pluse labeled with BrdU after exposure to ATRi for 24 h and chased for another 24 h with ATRi), at 24 h post-stimulation (red) until collection (with ATRi for 30 min, then chased with ATRi during 24–48 h) or for an hour (green) (ATRi for 30 min and then chased without ATRi during the 24–48 h). Flow cytometry profiles are shown with quantification in Supplementary Fig. 3d. **f** Cells were collected at 24, 30, 36, and 48 h post-stimulation and analyzed for Histone H3 S10 and PI. **g** Cells pulse-labeled with BrdU for 30 min at 10 h post-stimulation. G1 and S gates are shown. **h** Cells were collected for alkaline comet assay at 0 h or at 10 h post-stimulation. One representative of two independent experiments is shown. The alkaline tail moment is reported in arbitrary units (a.u.). Data are presented as mean values ±SD and statistical analysis used the two-tailed Mann−Whitney test. Source data are provided as a Source Data file.

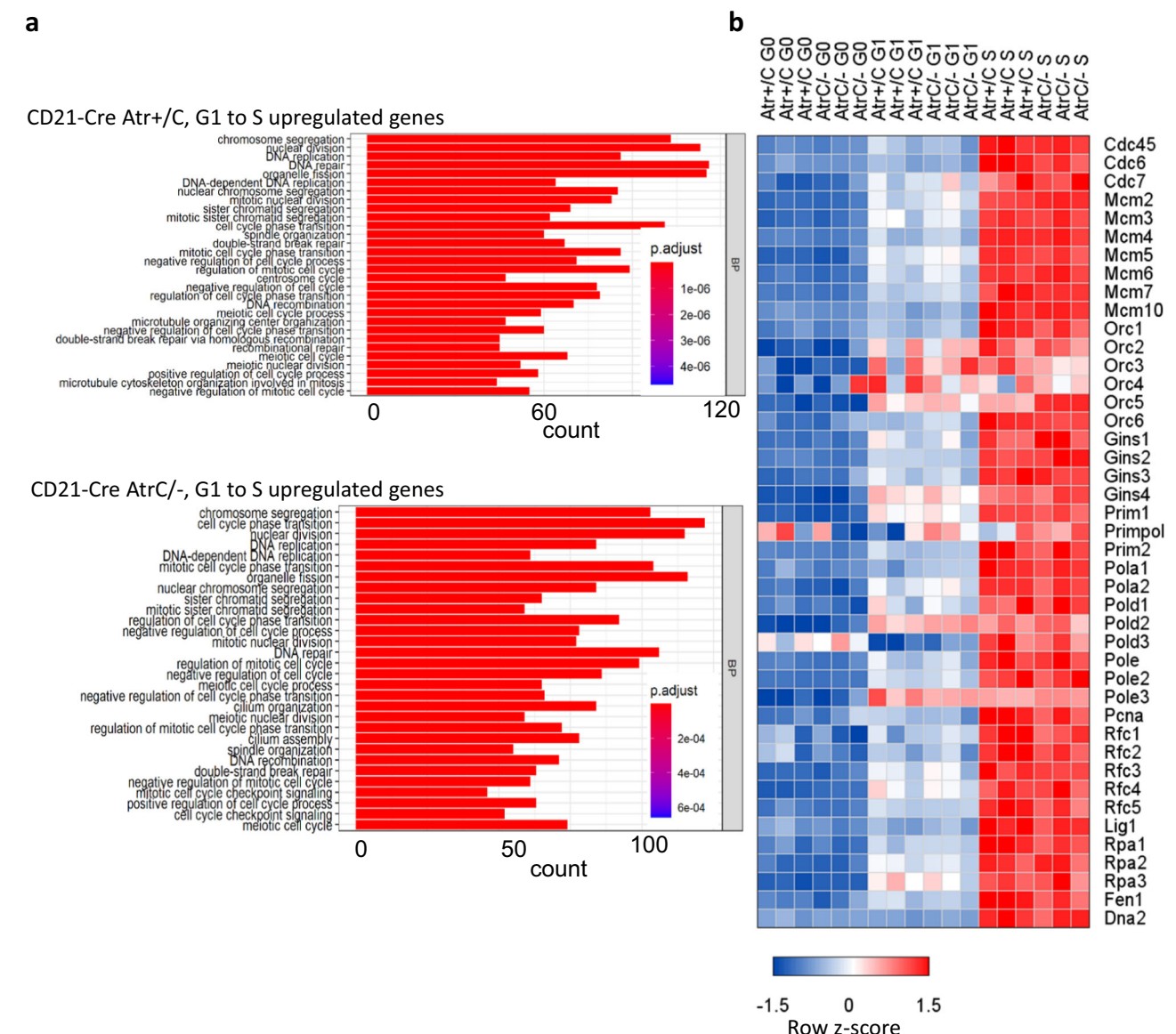

**Fig. 4 | ATR kinase does not affect the S phase transcriptional program. a** RNA-seq analyses from three independent culture splenic B cells from *CD21-Cre⁺ Atr⁺/C* and *Atr^C/-* mice at G0 (*t* = 0 h), G1 (*t* = 10 h), and S (*t* = 24 h) after IL-4 and anti-CD40 stimulations. The RNA-seq data were processed with DeSeq2 package, which uses a Wald test: the shrunken estimate of LFC is divided by its standard error, resulting in a z-statistic, which is compared to a standard normal distribution. Gene ontology (GO) analysis was performed on genes that are significantly upregulated from the G1 to S phase with a fold increase (FC) of at least 1.5 (log2 > 0.585) and an adjust *p* value < 0.01. Two tailed test was used. **b** A heatmap of the row z-scores of transcripts per million (TPM) of DNA replication genes in the transition from G0 to S phase is shown. Source data are provided as a Source Data file.

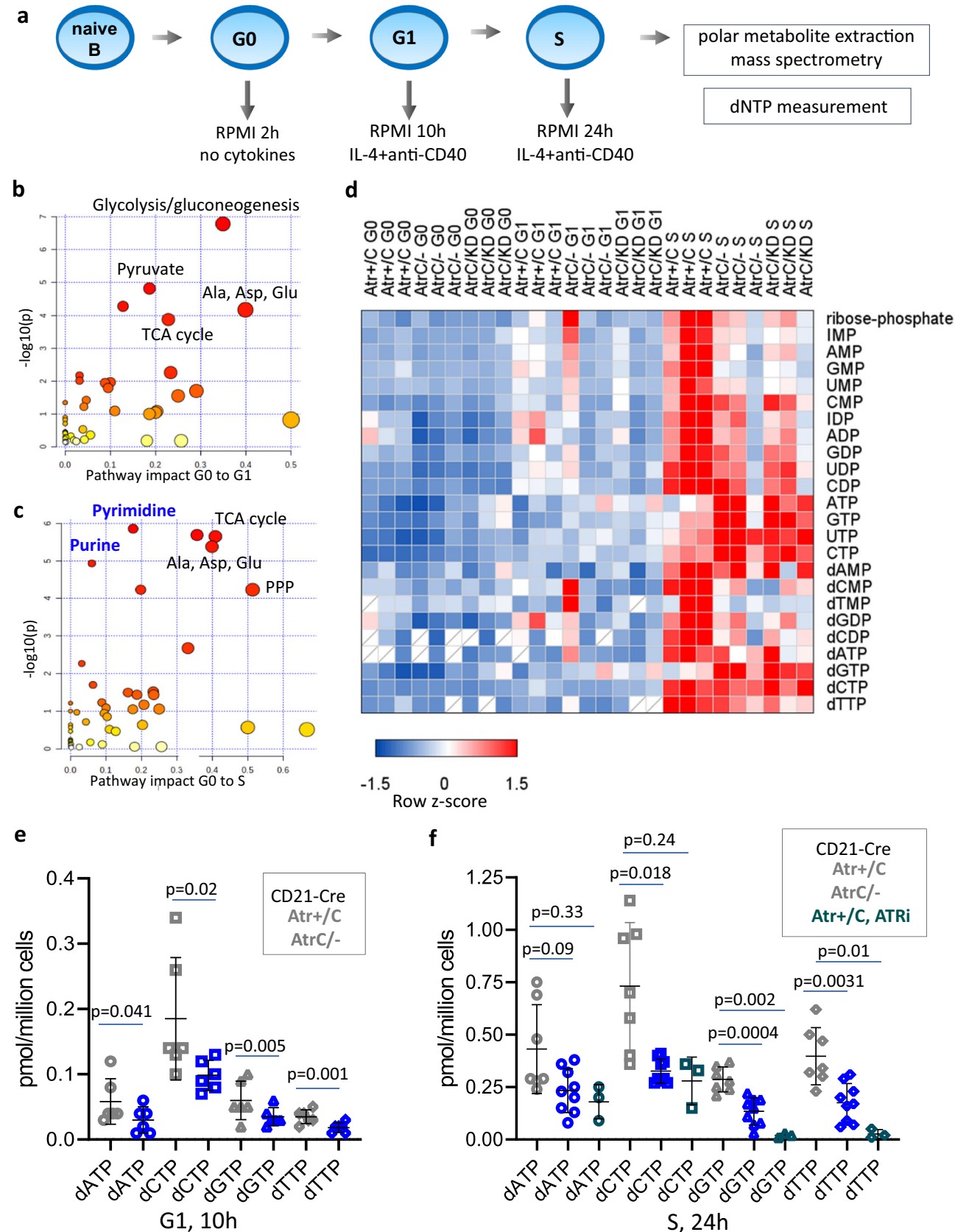

under low-dose ATRi conditions (IC20)[57,58], we chose IC90 (90% killing) to mimic complete genetic deletion of Atr and thereby identify pathways that support the essential ATR function in normal proliferation. Using the FDR < 0.2 as a cutoff, the chance of a random hit appearing in

more than one screen is lower than 5%. Previous DNA damage response screens, including those using ATR inhibitors, also used a cutoff of 0.2 when common targets from multiple independent screens are compared[57–60]. With this standard, 57 gRNA-targeted genes were

**Fig. 5 | Metabolome profiling and direct dNTP measurement reveal selective nucleoside and deoxyribonucleotide defects in *Atr*-deficient B cells.**
**a** Schematic representation of the time points collected (2 h, 10 h, and 24 h) for the metabolome and dNTP analyses. **b, c** Pathway impact analysis performed using a hypergeometric test on ctrl *CD21-Cre+ Atr+/C* B cells comparing the transition from G0 to G1 (**b**) and from G0 to S phase (**c**). **d** Heatmap of the row z-scores of integrated peak area values of several metabolites of the purine and pyrimidine pathways for *CD21-Cre+ Atr+/C*, *Atr^C/-*, and *Atr^C/KD* in G0, G1, and S phases. **e** dNTP (dATP,

dCTP, dGTP, and dTTP) amounts were quantified in pmol/million cells from six independent *CD21-Cre+ Atr+/C* or *Atr^C/-* B cells collected at 10 h post-activation. Statistical analysis was performed using a two-tailed t-test. **f** Analysis as in **e**, but cells were collected at 24 h post-activation from seven or nine independent samples. Three independent ATRi-treated *CD21-Cre+ Atr+/C* B cell cultures were added to the analysis. ATRi was added at 10 h post-stimulation and kept in culture until 24 h. Statistical analysis was performed using a two-tailed *t*-test. Source data are provided as a Source Data file.

enriched in the ATRi- or CHK1i-treated groups (compared to DMSO-treated control). GO analysis revealed that most of the enriched genes are implicated in either of two functional processes (Fig. 6f, g and Supplementary Fig. 6d, and Supplementary Data 1). One group is comprised of genes that facilitate S/G2/M cell cycle transitions and the mitotic checkpoint, such as Cdc25a and Cdc25b. Of note, this group was also observed previously in low-dose (IC20) ATRi-resistant screens designed to identity novel genetic vulnerabilities induced by ATR inhibition. Although the desired hits of those screens were gRNAs that are depleted upon ATRi treatment, a similar array of cell cycle transition and checkpoint genes were found to be enriched. Interestingly, the second, much larger, group of gRNAs represented genes implicated in various aspects of replication origin firing, and enriched uniquely in our high-dose ATRi/CHKi screens. This includes genes in the core origin firing complex (e.g., Cdc45, Cdc7, Dbf4, and Gins3), chromatin modulators implicated in late origin firing (Asf1b and Baz1a), and transcriptional factors licensing origin firing (E2f2, Ccne2, Ccna2). These findings suggest that ATR supports normal DNA replication through its ability to suppress origin firing, and that excessive origin firing in early S phase is responsible for the phenotype of Atr-deficient B cells, including the exhaustion of nucleotides and other replication factors.

## CDC7 and CDK1 concomitant inhibition rescues the productive DNA synthesis and nucleotide supply in ATR- deficient B cells

If the essential function of Atr in the early S phase is to prevent excessive origin firing, then it should be possible to rescue productive DNA replication in Atr-deficient primary B cells through partial suppression of origin firing. Indeed, several previous studies in human immortalized or cancer cells have suggested a function of ATR and CHK1 in controlling replication initiation and origin firing by counteracting CDC7 or CDK1 activties[38,61,62]. Given that factors implicated in origin firing appear in our CRISPR survival screen, we set out to test whether suppressing origin firing could restore productive replication in primary cells with clean deletion of ATR. To this end, we examined whether inhibition of CDC7, a kinase that promotes origin firing by phosphorylating the MCM helicase, rescues the phenotypes mentioned above. Thus, cells were treated with a CDC7 kinase inhibitor (XL413), either alone or together with the CDK1 inhibitor RO-3306. CDK1 inhibitor was included because premature activation of the cyclin A-CDK1 complex promotes late origin firing[63], and we observed premature mitosis downstream of CDK1 in Atr-deficient cells (Fig. 3f and Supplementary Fig. 3d). Although CDC7i treatment alone enhanced BrdU labeling intensity in *Atr*-deficient cells, it was not sufficient to restore productive DNA replication, as measured by increased DNA content (Supplementary Fig. 7a), unless combined with CDK1 inhibition (Fig. 7a). Notably, CDK1 and CDC7 inhibitors need to be added before DNA replication – at G1 (10 h) to reduce BrdU negative frequency of S phase *CD21-Cre+Atr^C/-* cells and partially but significantly restore effective DNA replication to mid and late S phase (Fig. 7a–c, and Supplementary Fig. 7b, c). Correspondingly, dual CDK1 and CDC7 inhibition also reduced DNA damage response in *CD21-Cre+Atr^C/-* cells, including phospho-KAP1, phospho-RPA, and γH2AX (Fig. 7d). Moreover, treatment reduced DNA strand breaks (Fig. 7e) and partially restored fork speed (Fig. 7f) in *Atr-deficient* cells. Notably, CDK1 and

CDC7 inhibitors also increased fork speed in *Atr-proficient* control cells, suggesting that even in the presence of ATR, the speed of the individual replication fork is limited by the number of replication forks firing at the same time. Consistent with overconsumption as the cause of nucleotide depletion, the dNTPs (dATP, dGTP, dTTP) in *CD21-Cre+Atr^C/-* cells were also partially restored upon concomitant CDC7i and CDK1i inhibition (Fig. 7g and Supplementary Fig. 7d). RNA-seq analyses also revealed significant attenuation of DNA damage response and p53 signatures in *CD21-Cre+Atr^C/-* cells following CDK1 and CDC7 inhibition (Supplementary Fig. 7e). Thus, our data suggest that ATR promotes productive DNA replication in activated lymphocytes by spacing out the replication origin firing to prevent the exhaustion of replication proteins and nucleotides. Therefore, in mid- or late-S phase cells with limited residual origins, ATR inhibition does not prevent robust replication.

## Discussion

Much is now known about how ATR coordinates the cellular response to DNA replication stress induced by various genotoxins and activated oncogenes in human cancer cell lines[3]. However, the mechanisms underlying its essential role during unperturbed DNA replication of mammalian cells remain elusive, in part due to the challenge of maintaining ATR-deficient cells in culture. Most somatic cells of mammals, including peripheral lymphocytes, reside in a resting G0 state. Here we show that loss of ATR is well-tolerated in peripheral naïve B cells, providing a genetic model in which to study the essential functions of ATR in normal replication. Upon cytokine stimulation, splenic naïve B cells efficiently exit G0, progress through the G1-S transition, and begin DNA replication in early S phase. Using this system, we show that Atr-null or Atr-KD (kinase-dead) B cells initiate DNA replication robustly and express a normal S phase transcriptome. However, productive DNA replication, as defined by increased overall DNA content, is aborted, such that ~50% of the mid-S phase cells fail to incorporate BrdU during a 30-min labeling period. These Atr-deficient cells also display markers of the DNA damage response indicative of heightened stalling and collapse of DNA replication forks. Moreover, since both null and kinase-dead Atr mutations yield similar replication defects, our results imply that the kinase activity of ATR is necessary for productive DNA replication. This notion is consistent with our previous observations that kinase-dead ATR can inhibit the dynamic exchange of wildtype ATR and RPA on ssDNA in a dominant-negative manner, leading to replicative stress at difficult and late replication regions, ribosomal DNA, and telomeres[64]. Importantly, the replication defects of Atr-deficient cells can be recapitulated in normal Atr-proficient B cells by treatment with a pharmacological ATR inhibitor during G1 but not during mid/late S phase, indicating that ATR kinase activity is dispensable for ongoing replication and is most essential in first early S phase, after the G0 arrested primary cells entering the cell cycle. Similar phenotypes have been reported in human cancer cells with ATR kinase inhibitors or under replication stress. This phenotype is not generated by inhibitors of the other damage-inducible PI3KKs (i.e., ATM or DNA-PK), highlighting a unique function of ATR in normal replication. Together, our findings indicate that Atr kinase activity is specifically required during the early S phase for productive DNA replication in normal mammalian cells. This requirement seems most

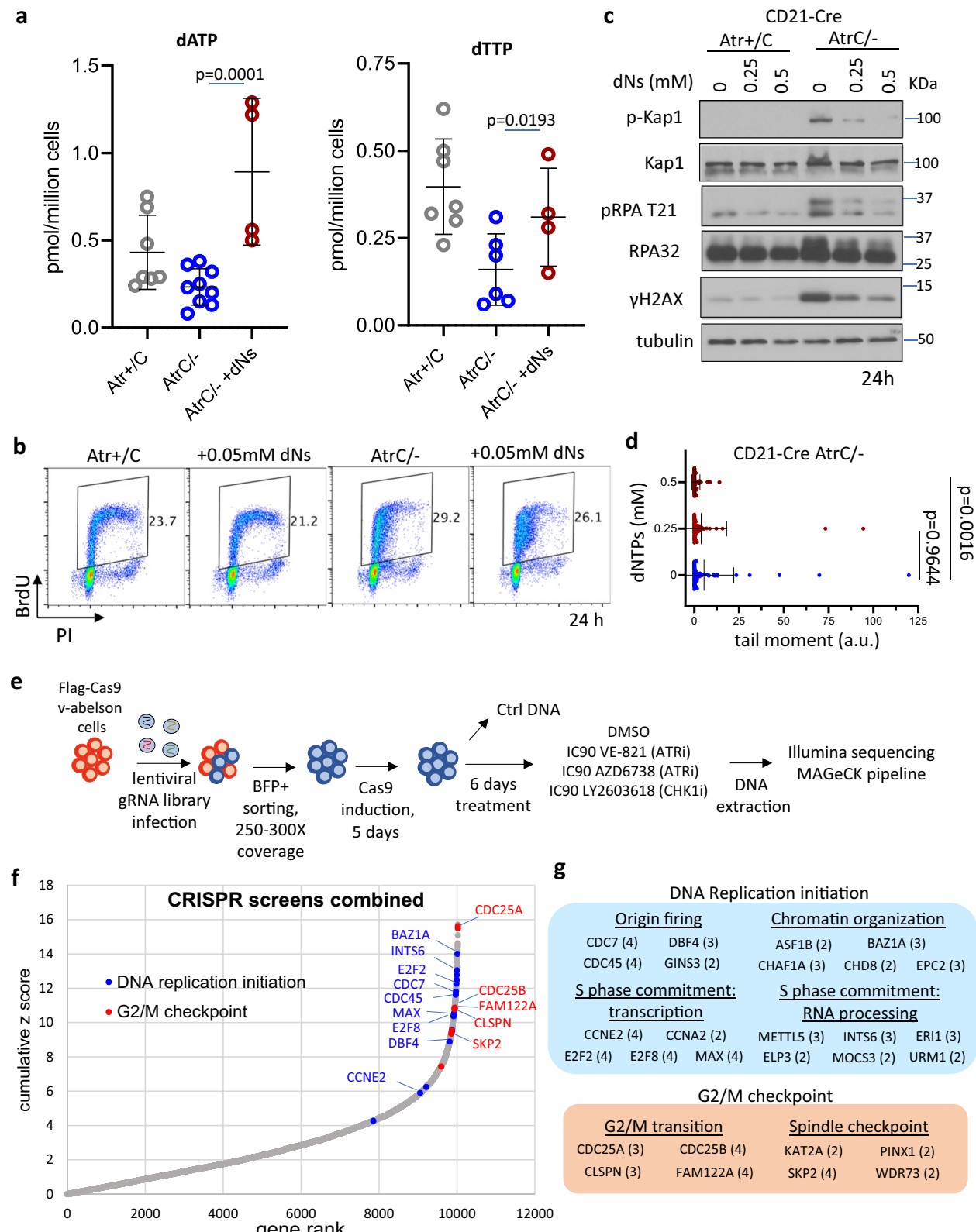

**a** dATP / dTTP plots, **b** BrdU/PI flow cytometry, **c** CD21-Cre immunoblots, **d** tail moment plot, **e** CRISPR screen workflow, **f** CRISPR screens combined, **g** gene category boxes.

prominent in primary cells that enter the cell cycle from G0 phase and is especially important for the very first cell cycle when the resources are limited.

How does ATR function in early S phase to support productive DNA replication and proliferation of unstressed mammalian cells? Interestingly, studies of budding yeast have shown that the steady-state dNTP levels of G1 cells are insufficient to support processive DNA

synthesis after the G1/S transition. However, with the onset of DNA replication, the yeast Mec1[ATR] kinase becomes transiently activated by ssDNA structures that arise during early S phase[7]. Upon activation, Mec1[ATR] and its downstream effector kinases, Rad53[CHK1] and Dun1, promote de novo dNTP synthesis by increasing the activity of ribonucleotide reductase (RNR), largely through degradation of the RNR inhibitor Sml1[8,10]. However, since orthologs of Sml1 or Dun1 have not

**Fig. 6 | Effects of deoxyribonucleoside supplementation and CRISPR-Cas9 screen in v-abl virus-transformed B cells. a** dNTP analysis was performed in *CD21-Cre⁺ Atr⁺/C* and *Atr^C/-* B cells collected at 24 h post-stimulation (same data as in Fig. 5f) and in *CD21-Cre⁺ Atr^C/-* B cells supplemented with 2.5 mM of deoxyribonucleosides for 14 h (added at 10 h). Quantifications of dATP and dTTP are reported. Statistical analysis used two-tailed *t*-test. **b** *CD21-Cre⁺ Atr⁺/C* and *Atr^C/-* B cells were treated at 10 h post-stimulation with 0.05 mM of deoxyribonucleosides. Samples were pulse-labeled with BrdU for 30 min at 24 h. **c** *CD21-Cre⁺ Atr⁺/C* and *Atr^C/-* B cells were cultured with 0 mM, 0.25 mM, or 0.5 mM of deoxyribonucleosides supplemented at 10 h post-stimulation. Cells were collected at 24 h, and western blots were performed for the proteins indicated. **d** Alkaline comet assay was performed at 24 h post-stimulation in *CD21-Cre⁺ Atr^C/-* cells untreated or treated with 0.25 mM or 0.5 mM of deoxyribonucleosides. One representative of two independent experiments is shown. Data are presented as mean values ±SD. Statistical analysis done with the two-tailed Mann–Whitney test. **e** Flag-Cas9 v-abl B cells were infected with a BFP-expressing lentiviral gRNA library. BFP+ cells were sorted and Cas9 induced with doxycycline for 5 days. Cells were treated with the indicated inhibitors for 6 days, after which DNA was collected for sequencing. Details are described in the methods. **f** The cumulative Z score was calculated from the four screens combined, and genes were ranked accordingly. Some of the DNA replication initiation and G2/M checkpoint genes with an FDR < 0.2 in at least two of the four screens have been highlighted in the gene rank plot. The CRISPR data were analzyed via the MAGeCK package and the FDR and Z score are direct output from the MAGeCK pipeline. **g** The DNA replication initiation and G2/M checkpoint genes with an FDR < 0.2 in at least two of the four screens are grouped. Genes have been divided into sub-categories based on biological processes. The numbers in brackets represent the number of screens in which each gene has been significantly picked. Source data are provided as a Source Data file.

been found in mammals, this pathway cannot account for the essential function of Atr in normal proliferation of mammalian cells. Here we show that the dNTP levels of cytokine-induced Atr-proficient B cells are dramatically increased at the G1/S transition. In ATR-deficient cells, the dNTP levels also increase at the G1/S transition, but they are markedly lower than those of control cells during both G1 and S phase. Nonetheless, despite relatively low dNTP levels, ATR-deficient B cells initiate DNA replication robustly, as measured by both the intensity and frequency of BrdU incorporation during the early S phase (Fig. 3a and Supplementary Fig. 3a). However, by the mid-S phase, roughly half of these cells undergo extensive stalling and collapse of replication forks. Unlike in budding yeast[7], we were unable to restore productive DNA replication in Atr-deficient B cells by replenishing cellular dNTP levels. Like others, we also noted less induction of RNR in the *Atr-deficient* cells[53]. Taking advantage of the activity of dCK[54] in lymphocytes, we supplemented the cells with dNs that bypass RNR and successfully increased dNTP concentration in the cells. The dN supplementation failed to restore DNA replication but significantly attenuated the DNA damage responses. The concentration of the dN supplementation did not affect replication in control cells, suggesting the lack of replication rescue cannot be explained by nucleotide imbalance from the supplementation. RNA-seq analyses of these cells further confirmed that the replication-associated transcriptional program is robustly induced, and nucleotide biogenesis pathways are normally expressed.

Our inability to rescue productive DNA replication in Atr-deficient B cells by effectively increasing cellular dNTP implied that additional replication factors are also limited in these cells. Therefore, to identify a mechanism that would overcome these deficits, we conducted CRISPR-Cas9 screens using near-lethal doses of ATR and CHK1 inhibitors. We reasoned that these factors, as well as the underlying pathways that mediate Atr's essential function, would be enhanced in cells that survive severe ATR/CHK1 inhibition. In these screens, we observed an enrichment for two distinct classes of genes, one comprised of factors traditionally associated with the G2/M and spindle checkpoints (e.g., Cdc25a, Cdc25b) and another comprised of proteins implicated in various aspects of replication initiation, as described below. Of note, CRISPR-Cas9 screens using low-dose ATRi (IC20) were previously conducted to identify gRNA-targets that would represent genetic vulnerabilities toward ATR inhibition, and as such might serve as valuable synergistic targets for ATRi-based therapies[57,65]. Interestingly, those low-dose ATRi screens also found an enrichment of gRNAs for the G2/M and spindle checkpoints factors, but not of gRNAs for replication initiation factors. In that setting, depletion of Cdc25a was thought to prevent premature mitosis and the consequent mitotic catastrophe caused by ATR and CHK1 inhibition. As discussed below, gRNAs for Cdc25 phosphatase may have been enriched in our high-dose ATR

inhibitor screens due to Cdc25's ability to activate CyclinA-CDK1, which promotes late replication origin firing[63].

The largest class of gRNAs recovered in our high-dose ATRi screens is comprised of factors known to promote various aspects of origin firing, including core origin launching factors (*e.g.*, Cdc45, Cdc7, Dbf4, Gins3), chromatin remodeling for replication origin (e.g., Asf1b, Baz1a), and transcription factors for replication initiation (e.g., E2f2, Ccne2, Ccna2). The unique enrichment of gRNAs for replication initiation factors suggests that ATR-mediated suppression of origin firing is a central aspect of ATR essential function in normal DNA replication. To test this hypothesis, we sought to partially suppress origin firing in Atr-deficient cells using a combination of CDC7 and CDK1 inhibitors. The CDC7 kinase, which phosphorylates the MCM helicase complex, is required for origin firing in general[66], while hyperactivation of CyclinA-CDK1 was previously found to promote late origin firing in the absence of ATR or CHK1[63]. In addition, a recent study has shown that CDK1 is also activated at the G1/S transition, where it acts both collaboratively and redundantly with CDC7 to drive origin firing by phosphorylating the MCM helicase complex[67]. Our observation that co-inhibition of both kinases rescues productive DNA replication in Atr-deficient cells, indicates that the suppression of origin firing is a critical aspect of Atr's essential function. Moreover, CDC7 and CDK1 co-inhibition also restored dNTP levels in these cells, consistent with the notion that dNTP levels in Atr-deficient S phase cells are reduced due to excessive origin firing and consequent overuse of dNTP pools. CDC7 inhibition alone partially suppressed the DNA damage response but was not sufficient to restore productive replication. Thus, in addition to suppressing origin firing, CDK1 inhibition may also prevent premature mitosis and the activation of structure-specific nucleases that can collapse stalled replication forks[68] and shutdown RNR expression[69]. In light of these results, we propose that ATR ensures productive DNA replication and normal cell proliferation by preventing excessive origin firing and the exhaustion of dNTPs and other replication proteins. At any given time, the availability of key replication factors, including proteins and dNTPs, should reflect a balance between their production and consumption. Given the large genome of mammalian cells, it may be more efficient to space out the firing of individual origins during early S phase to allow reuse of some replication factors and temper the demand for dNTPs. Thus, cells may be more sensitive to ATR inhibition during early S phase, when late-replicating origins must be repressed, than during mid- or late-S phase, when fewer origins remain. A similar mechanism might underlie the critical role of ATR in sensing deoxynucleoside imbalance[55]. Consistent with this model, primary cells entering the cell cycle from G0 (in the first 24 h) are most sensitive to ATR inhibition/deletion in comparison to cell lines in continuous culture or later cell cycles, which could benefit from the significant levels of replication factors left from previous cell divisions.

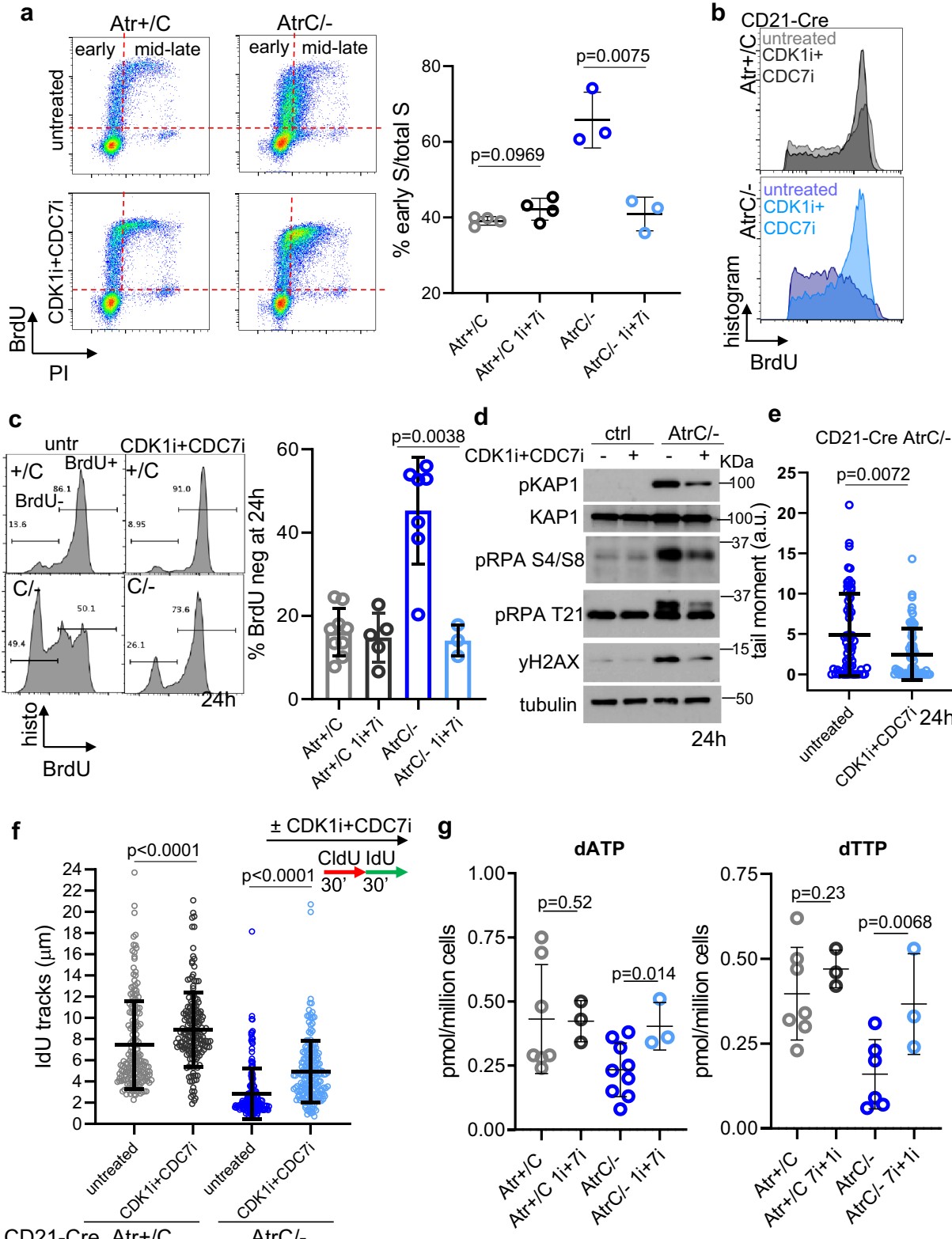

Whether this suggests that primary and stem cells might be more sensitive to ATR inhibition than cancer cells requires further investigation. In this context, cancer cells often suffer from oncogenic replication stresses that enhance their sensitivity to ATR inhibition.

A large body of work has identified at least four well-defined mechanisms by which ATR promotes the cellular response to replication stress: suppression of origin firing, activation of cell cycle checkpoints, stabilization of stalled forks, and induction of fork restart[3]. It is feasible that one or more of these mechanisms may also mediate the essential function of ATR in the proliferation of unstressed cells. The identification of ATR's common and unique roles in tumor cells vs. normal cells would provide the first step to identify tumor-specific vulnerability and minimize the toxicity of ATR inhibition. Our

**Fig. 7 | CDC7 and CDK1 concomitant inhibition rescues the productive DNA synthesis and nucleotide supply in Atr-deficient B cells. a** *CD21-Cre*[+] *Atr*[+/C] and *Atr*[C/-] cells untreated or treated at 10 h post-stimulation with 2.5 μM of CDC7i (XL413) and 2.5 μM of CDK1i (Ro-3306) were pulse-labeled with BrdU for 30 min and collected at 24 h. Representative flow cytometry dot plots are reported. Grid indicates early vs. mid-late S phase cells. Quantification of early S phase and mid +late S phase cells from three or four biologically independent experiments. Data are presented as mean values ±SD. Statistical analysis used two-tailed t-test. **b** Overlay of cell cycle histograms of BrdU-positive cells with or without CDC7i + CDK1i for *CD21-Cre*[+] *Atr*[+/C] and *Atr*[C/-] cells. **c** The percentages of BrdU negative and positive S phase cells at 24 h from several independent experiments of *CD21-Cre*[+] *Atr*[+/C] and *Atr*[C/-] cells untreated or treated with both CDC7i and CDK1i for 14 h. **d** Indicated cells were left untreated or treated with both CDC7i and CDK1i and

proteins collected at 24 h. **e** Alkaline comet assay was performed in cells untreated or treated with both CDC7i and CDK1i for 14 h, and the tail moment of single cells is shown in arbitrary units. One representative of two independent experiments is shown. Data are presented as mean values ±SD. Two-tailed Mann-Whitney test was used. **f** *CD21-Cre*[+] *Atr*[+/C] and *Atr*[C/-] cells were left untreated or treated with both CDC7i and CDK1i for 14 h. In all conditions, cells were pulse-labeled at 24 h with CldU for 30 min and then with IdU for additional 30 min. The length of IdU fibers was plotted, one representative of two independent experiments is shown, data are presented as mean values ±SD and the two-tailed Mann−Whitney test was used. **g** *CD21-Cre*[+] *Atr*[+/C] and *Atr*[C/-] cells were left untreated or treated with both CDC7i and CDK1i for 14 h. Samples were collected at 24 h for dNTP measurement. Untreated samples are from Fig. 5f. Quantifications of dATP and dTTP are reported. Two-tailed *t*-test was used. Source data are provided as a Source Data file.

data indicate that ATR-mediated suppression of origin firing during early S phase is required to support productive DNA replication in normal lymphocytes that transition from G0. As in budding yeast, we found that ATR[Mec1] is not necessary for the initiation of DNA replication in cytokine-induced B cells. In addition, ongoing replication was not abrogated by ATR inhibition beginning in mid- or late-S phase. Thus, although ATR and its kinase activity facilitate DNA replication by preventing overuse of limited replication factors, they appear to be dispensable for the actual mechanics of fork initiation and progression. Indeed, recent reconstitution of fast and efficient human DNA replication[70] does not include ATR kinase. Accordingly, the essential role of ATR in normal replication is most prominent in early S phase, when cells maintain a large number of dormant origins which, if fired inappropriately, would readily deplete available pools of replication factors. In contrast, transient ATR inhibition during mid or late S phase does not block productive replication, presumably because these cells possess a relatively smaller reserve of dormant origins or have accumulated significant replication factors. This stage-specific effect of ATR on productive DNA replication may provide a means to dose ATR inhibitors to minimize normal tissue toxicity during cancer therapy.

## Methods

### Mice and cell lines

Mice used in this study were either in 129/Sv, C57BL/6, or mixed background. The *Mb1*[Cre] (*CD79a*) knockin allele[29], *CD21-Cre* transgene allele[30] and the Rosa-ER-Cre alelle[71] were described previously. We generated the *Atr* kinase-dead (*Atr*[KD]) allele that expresses a catalytically inactive ATR protein[64]. Dr. Eric Brown generously shared the *Atr* conditional allele (*Atr*[C])[5] and the knockout allele (*Atr*[KO])[4]. In the *Atr*[KO] allele, exons 1–3 of the *Atr* gene are replaced by a Neo-resistant gene[4]. After Cre expression, the *Atr*[C] will be converted to *Atr*[Δ] (or del) allele, in which the two exons near the C-terminus that encodes the Atr kinase domain are removed, and a subsequent frameshift is generated[5]. Since the *Atr*[C/Δ] and *Atr*[KO] involve distinct exons of the *Atr* gene, the PCR genotyping designed for *Atr*[C/Δ] would show a "WT" band in *Atr*[KO] cells. Similarly, PCR genotyping designed for *Atr*[KO] would detect the intact exon 1–3 in the *Atr*[C/Δ] mice. This can be seen in Supplementary Fig. 1k where the v-abl kinase transformed B line 8412 is *Atr*[C/KO]. In both cases, the ATR protein cannot be detected after deletion[4,5], resulting in a complete loss of ATR. In our studies, we used the *Atr*[KO] and *Atr*[Δ] alleles interchangeably and referred to them as *Atr*[-] together for simplicity. v-abl kinase transformed B cell lines were generated using the bone marrow derived from the femurs of 3–4 weeks old mice of the indicated genotypes. Independent infections of bone marrow cells were performed with v-abl retrovirus and cells were cultured for ~6 weeks until viable dominant transformed clones were generated. Human cell lines 293T and RPE cell lysate were used in Supplementary Fig. 1b as human cell comparison. All animal work was conducted in the specific

pathogen-free facility at Columbia University Medical Center and was approved by the Institutional Animal Care and Use Committee (IACUC) of Columbia University.

### B cell development analyses by flow cytometry

B cell developmental analyses were carried out in 8–12-weeks-old mice. Single-cell suspensions were prepared from the bone marrow (BM), and spleen (Spl) of the mice of the indicated genotypes, and ~1 × 10[5] cells in 1X PBS were stained for 15 min at RT using fluorescence-conjugated antibodies. Cells were analyzed by flow cytometry on a FACSCalibur flow cytometer (BD Biosciences), and data were processed using the FlowJo software package. The following antibody cocktail was used for B cell analyses, with each antibody used at 1:200 dilution: FITC anti-mouse CD43 (Biolegend, 553270); PE anti-mouse IgM (Southern Biotech, 1020-09); PE-Cyanine5 anti-human/mouse CD45R (B220), (eBioScience, 15-0452-83); and APC anti-mouse TER119 (Biolegend, 116212). Live cells were gated based on forward, and side scatters (FSC and SSC, respectively). Red blood cells (RBC) were excluded using a Ter119 antibody specific for RBC.

### Apoptosis staining

Primary B cells undergoing CSR were collected at 0 h, 24 h, 48 h, 72 h, and 96 h for apoptosis assay. Cells were resuspended in 100 μl 1x Annexin binding buffer (10 mM Hepes adjusted to pH7.4, 140 mM NaCl, and 2.5 mM CaCl2) at a concentration of ~10[6] cells/ml, with 1 μl 100 μg/ml PI and 1 μl FITC-Annexin V (BD Pharmingen, Cat. No. 556419) antibody. After incubation for 15 min at room temperature in the dark, an additional 400 μl 1X Annexin binding buffer was added into each tube. Cells were analyzed on the flow cytometer (Attune NxT cytometer, ThermoFisher Scientific) immediately.

### Immunization, germinal center analyses, and immunohistochemistry

*CD21-Cre*+ *Atr*[+/C], and *Atr*[C/-] mice were immunized intraperitoneally (i.p.) with 1.5 × 10[8] sheep red blood cells (SRBC) (Cocalico Biologicals, Lot No CBI29125) in 1X PBS at day 1 and day 5. At day 14, splenic B cells were collected and stained with FITC anti-mouse/human GL7 (Biolegend, Cat. No. 144604) and PE hamster anti-mouse CD95 (BD, Cat. No. 561985) for the following flow cytometry analysis. Small pieces of the spleen were formalin-fixed and sectioned at 5 μm for standard immunohistochemistry staining. Molecular Pathology shared resource (MPSR) at Herbert Irving Comprehensive Cancer Center (HICCC) provided histology support.

### Class switch recombination and cell trace violet staining

For the Class Switch Recombination assay, single-cell suspensions of spleen cells were sorted with CD43 magnetic beads (MACS, Miltenyi), and CD43- B cells were cultured at a density of 5 × 10[5] cells per ml in RPMI medium supplemented with 15% FBS and 25 ng ml[-1] of IL-4 (R&D) and anti-CD40 (BD Biosciences). Cultured cells were maintained daily

at a density of $0.5-1 \times 10^6$ cells per ml. Cells were collected every day up to day 4 and were stained with FITC-conjugated IgG1 (1:200, A85-1, BD Pharmigen, 553443) and PE-Cyanine5-conjugated B220 (1:200, RA3-6B2, BD Pharmigen, 553091). Flow cytometry was performed on a FACSCalibur flow cytometer (BD Biosciences) or Attune NxT cytometer (ThermoFisher Scientific). Data obtained using CellQuest Pro or Attune NxT Software v4.2.0 were processed using the FlowJo software package. The proliferation of primary B cells undergoing CSR was analyzed using the Cell Trace Violet (CTV) kit (ThermoFisher Scientific, C34557) according to the manufacturer's protocol. Flow cytometry was performed on BD LSRII flow cytometer, and data were processed using FlowJo software package.

## BrdU cell cycle analysis

For BrdU/PI cell cycle analyses, B cells were incubated with 60 μM of 5-Bromo-2'-deoxyuridine (BrdU, Sigma) as indicated in the figure legends. Cells were fixed in cold 70% ethanol overnight, permeabilized, and DNA denatured with an acid solution (2 N HCl, 0.5% Triton X-100) for 30 min at RT and washed with sodium phosphate citrate buffer pH 7.4. Cells were then stained with FITC-conjugated mouse anti-BrdU antibody (1:10, BD Pharmingen, 556028) for 30 min. Finally, cells were incubated with Propidium Iodide (Sigma) and RNase A (Sigma) for an additional 30 min at RT in the dark. More than 50,000 cells/sample were acquired either on a FACSCalibur flow cytometer (BD Biosciences) or Attune NxT cytometer (ThermoFisher Scientific) and analyzed using the FlowJo software package.

## γH2AX flow cytometry

About 5 million cells/sample were harvested, washed with 1X PBS, and fixed with 300 μl of ice-cold methanol for 20 min at −20 °C. After centrifugation, cells were washed twice with 1X PBS. Cells were blocked with 5 mg/ml BSA in 1X PBS for 30 min and then incubated with primary antibody phospho-histone H2AX Ser139 20E3 (Cell Signaling #9718, 1:500) for 2 h at room temperature. After washing with 5 mg/ml BSA in 1X PBS, cells were incubated with secondary antibody (goat anti-rabbit Alexa488, Thermo Fisher Scientific, A-11008, 1:300) for 1 h at room temperature in the dark. Cells were washed with 5 mg/ml BSA in 1X PBS, centrifuged, and stained with the PI solution for 30 min at RT in the dark. For the PI solution, PI and RNase were diluted in 5 mg/ml BSA in 1X PBS to final concentrations of 20 μg/ml and 100 μg/ml, respectively. About 50,000 cells/samples were analyzed by FACS on a FACSCalibur flow cytometer (BD Biosciences) using FL1 (γH2AX) and FL3 (PI) channels.

## Cellular dNTP measurement

Intracellular dNTPs were measured by HIV-1 RT-based dNTP assay as previously described[52]. Cell pellets were prepared from $2 \times 10^6$ primary B cells collected at the time of purification (G0), 10 h (G1), or 24 h (S) post-stimulation with IL-4 and anti-CD40. Cells were washed with 1X PBS, and PBS was removed as much as possible. The cell pellet was lysed by quickly resuspending the cells in ice-cold 65% methanol (100 μl per $1 \times 10^6$ cells). Samples were vigorously vortexed for 2 min. Cells were completely lysed by incubating at 95 °C for 3 min and then chilled on ice for one minute. The Eppendorf tubes were centrifuged for 3 min at 5000×$g$, and the 65% methanol solution was transferred to a new labeled tube with identifiable numbers. The tubes with the pellet were discarded. Samples were dried by speed vacuum and stored at −80 °C until processing. Dried samples were resuspended and diluted to be within the linear range of the assay. A 5′ $^{32}$P-end-labeled 18-mer DNA primer (5′-GTCCCTCTTCGGGCGCCA-3′; Integrated DNA Technologies) was annealed to one of four 19-mer DNA templates (3′CAGGGAGAAGCCCGCGGTN-5′; Integrated DNA Technologies), where N represents a variable DNA nucleobase. Resuspended cellular extracts and purified HIV-1 HXB2 RT were used to extend the primer during incubation for 5 min at 37 °C.

Water or 0.5 mM dNTP mix was used instead of cellular extracts as a negative and positive control, respectively. Reactions were terminated with 10 μl 40 mM EDTA and 99% (v/v) formamide and denatured by incubation at 95 °C for 2 min. Reaction mixtures were resolved on a 14% urea-PAGE gel (AmericanBio, Inc.) and analyzed using a PharosFX molecular imager (Bio-Rad). Single-nucleotide extension products were quantified using Image Lab software, version 5.1.2 (Bio-Rad). Statistical analyses were performed using the unpaired Student's $t$ test.

## Polar metabolite extraction and mass spectrometry

Polar metabolite extraction was performed essentially as described in[50]. B cells cultured in the RPMI medium were incubated for 2 h in phenol red-free RPMI with dialyzed 10% FBS prior to metabolite collection. At the time points indicated, 6 million cells/sample were collected by centrifugation (500 g, 5 min) in 15 ml falcon tubes, and cells were immediately resuspended in 4 ml of 80% methanol stored in −80 °C. Cells were incubated at −80 °C for 15 min. The cell lysate/methanol mixture was centrifuged at 14,000×$g$ for 5 min in the cold room to pellet cell debris and proteins. The supernatant was transferred to 50 ml conical tubes on dry ice. 500 μl of 80% methanol (−80 °C) was added to the 15 ml tubes, and the pellet was resuspended by vortexing and pipetting up and down. The mixture was transferred to a 1.5 ml tube on dry ice. Tubes were spun in a microcentrifuge at full speed for 5 min in the cold room. The supernatant was transferred to the 50 ml conical tubes on dry ice collected above. 500 μl of 80% methanol (−80 °C) was added again to the 1.5 ml tubes, and the pellet was resuspended. After centrifugation at full speed for 5 min in the cold room, the supernatant was transferred to the 50 ml conical tubes on dry ice. After pooling the three extractions, the samples were equally divided into 1.5 ml tubes and completely dried by speed vacuum. Samples were stored at −80 °C until processing. Liquid chromatography and mass spectrometry were performed as described in ref. 50. Relative quantification and integrated peak areas based on total ion current were used for statistical analyses. MetaboAnalyst (https://www.metaboanalyst.ca/) web tool was used for Principal Component Analysis (PCA) and pathway analyses, for which the scatter plots were generated using a hypergeometric test as enrichment method, setting the FC > 1.5 (log2 > 0.585) and the $p$ value < 0.05. Heatmaps were generated using the integrated peak area values and the $z$-scores of the values in each row (subtract the mean and divide by the standard deviation).

## RNA isolation and RNA-sequencing

RNA was isolated from B cells using TRIzol (Invitrogen) following the manufacturer's protocol. RNA quality was determined using the Bioanalyzer RNA kit (Agilent), and samples with RNA Integrity Number (RIN) higher than 9 were sent for next-generation sequencing (NGS). Poly-A pull-down was used to enrich mRNAs from total RNA samples, then library construction was performed using Illumina TruSeq chemistry. Libraries were then sequenced using Illumina NovaSeq 6000 at Columbia Genome Center. Samples were multiplexed in each lane, which yields a targeted number of paired-end 100 bp reads for each sample. Real-Time Analysis (RTA, Illumina) was used for base calling and bcl2fastq2 (version 2.20) for converting BCL to fastq format, coupled with adaptor trimming. A pseudo alignment to a kallisto index created from transcriptomes (Mouse: GRCm38) was then performed using Kallisto (0.44.0). It was finally tested for differentially expressed genes under various conditions using Sleuth, an R package designed to compute transcript and gene-level differential expression from Kallisto abundance files. Heatmaps were generated using tpm (transcription per million) values and the z-scores of the values in each row (subtract the mean and divide by the standard deviation). When indicated, Pearson distance clustering was applied. Gene ontology (GO) analyses were performed using the R package ClusterProfiler[72,73],

and differentially expressed genes in which fold increase was greater than 1.5 (log2 > 0.585) with an adjusted $p$-value < 0.01 were considered. Gene Set Enrichment Analysis (GSEA) was performed using the Broad Institute software. Deseq2 package was used to compare gene expression. For significance testing, DESeq2 uses a Wald test: the shrunken estimate of LFC is divided by its standard error, resulting in a z-statistic, which is compared to a standard normal distribution.

## CRISPR-Cas9 Screen in murine v-abl kinase transformed B cells

A phenotypically WT (ROSA+/cre CTIPc/c Bcl2+) v-abl immortalized B cell line was infected with lentivirus for doxycycline-inducible SpCas9 (Addgene Plasmid #50661). Single clones were generated and screened for high SpCas9 inducibility after doxycycline treatment (Sigma Aldrich, D9891, 3 μg/ml for 3 days) via Western Blot and flow cytometry after intracellular staining using anti-Flag antibody (Sigma Aldrich, F3165). Once a suitable single clone was chosen, it was expanded and 100 million cells were infected with a BFP-expressing gRNA lentiviral library (Addgene Pooled Library #67988) at an infection rate of 25-30% to keep the Multiplicity of Infection (MOI) -1 and ensure 250–300x coverage of individual gRNAs. After expanding for 4 days, at least 35 million BFP+ cells were sorted to ensure 250–300x coverage. The sorted cells were grown to 280 million cells. Then, DNA from 3 samples of 30 million cells each were collected as "before doxycycline" controls, and the rest were treated with 3 μg/mL doxycycline for 5 days, refreshing the doxycycline-treated media every 2 days. On day 5, DNA from 3 samples of 30 million cells each was collected as "after doxycycline" controls, and the rest of the cells were divided into plates of 30 million cells each. Each IC90 drug treatment was determined by MTT assay under the same cell density and duration (ATRi VE821, 1.81 μM; ATRi AZD6738, 0.42 μM; CHK1i LY2603618, 0.4 μM) and a DMSO control (highest %v/v DMSO used in drug treatments) were performed in triplicate plates of cells and collected for DNA on day 6 after drug treatment. Cells were split or diluted as they became confluent, never going below 30 million cells in total. The IC90 at day 6 of treatment for each drug was determined by treating the inducible SpCas9 single clone (before gRNA library infection) with a dose curve of each drug in triplicate and taking an MTT assay measurement (Sigma Aldrich 11465007001) on day 6 following the manufacturer's instructions. The Hill coefficient was determined from the dose curve, which was then used to calculate the IC90. The gRNAs from the collected DNA samples were amplified and prepared for sequencing as previously described[74]. Briefly, the gRNA sequence from 250 μg of DNA from each sample was amplified using primers with Illumina sequencing adapters and a unique barcode for each sample, NEBNext High Fidelity Master Mix (NEB, M0541L), and 2.5 μg DNA for each reaction. The PCR products were purified using Zymospin V columns (Zymo Research, C1016) and then by Ampure XP beads (Beckman Coulter, A63880) according to the manufacturer's instructions using a 0.7x ratio. Samples were quantified via Bioanalyzer and qPCR using an Illumina library quantification kit (KAPA Biosystems, KK4824) according to the manufacturer's instructions. The libraries were pooled and sequenced on a NextSeq 550 machine (Illumina) using a High Output Kit v2.5 (75 Cycles, single-end, Illumina, Cat. No. 20024906) and 15% PhiX spike in. The results were analyzed via the MAGeCK and MAGeCK FluteMLE pipeline as detailed in the technical support page (https://sourceforge.net/p/mageck/wiki/Home/) using the DMSO samples as the baseline[57]. The $Z$-sore, $p$-value, and adjusted $p$-value (used as FDR) were direct output from the MAgeCK pileline and are always two-sided. Since each gene is targeted by 4–6 gRNA, the gene test and ranking used the modified robust rank aggregation (α-RRA) detailed in the original publication[57] and the software website (https://bitbucket.org/liulab/mageck/src/master/). For the analyses here, we used version 0.5.9. Genes with positive z-score and FDR < 0.2 identified in at least two independent screens out of 4 CRISPR screens

(three with ATR inhibitors and one with CHK1 inhibitors) were analyzed for Gene Ontology (GO) using R studio.

## Protein extraction, antibodies used, and western blot

Whole-cell extracts (WCE) were prepared using RIPA buffer (150 mM NaCl, 10 mM Tris-HCl pH 7.4, 0.1% SDS, 0.1% Triton X-100, 1% Sodium Deoxycholate, 5 mM EDTA) supplemented with 2 mM PMSF, 10 mM NaF, 10 mM β-glycerophosphate and protease inhibitor cocktail (Roche). Alternatively, cells were lysed in 137 mM NaCl, 50 mM Tris-HCl pH = 7.3, 10% Glycerol, 1% Triton X-100, 0.2% Sarcosyl, 10 mM NaF, and protease inhibitor cocktail (Roche), with the addition of Benzonase (Millipore) right before cell lysis. SDS-PAGE and western blots were performed following standard protocols. Primary antibodies used in the study are: ATR(1: 1000, Cell signaling #2790, Lot.7), pKAP1 S824 (1:1000, A300-767A, Bethyl Laboratories), KAP1 (1:1000, TIF1β, C42G12 Cell Signaling, #4124), pCHK1 Ser245 (1:1000, 133D3 Cell Signaling, #2348), CHK1 (1:1000, 2G1D5 Cell Signaling, #2360), pRPA32 T21 (1:10,000, Abcam, ab109394), pRPA S4/S8 (1:3000, A300-245A, Bethyl Laboratories), RPA32 (1:10,000, A300-244A, Bethyl Laboratories), pH2AX Ser139 (1:1000, 07-164, Millipore), vinculin (1:1000, V284, 05-386 Millipore), α-tubulin (1:1000, CP06, Calbiochem), pMCM2 S40 (1:1000, Abcam, ab133243). Western blot in Supplementary Fig. 3f was quantified using ImageJ. Briefly, CHK1 and vinculin protein bands were selected, and the relative density of the peaks was calculated as the area under the curve (arbitrary units). Finally, the ratio CHK1/vinculin was measured by dividing the area under the curve of CHK1 by the area under the curve of vinculin.

## Comet assay

Alkaline and neutral comet assays were performed following manufacturer instructions (Trevigen). Comets were acquired using a Nikon 80i fluorescence microscope equipped with a remote focus accessory and a CoolSNAP HQ2 camera unit with a Plan Apo 20X/0.75 objective. All images were processed with NIS-Elements AR software, and tail moments were calculated using the CometScore version 1.5 package. Statistical analyses were carried out using Graphpad Prism and Mann–Whitney non-parametric test.

## DNA fibers

B cell labeling was performed with 50 μM CldU (Sigma) for 30 min, followed by 250 μM IdU (Sigma) for an additional 30 min. Cells were harvested in ice-cold 1 x PBS, lysed using spreading buffer (0.5% SDS, 200 mM Tris-HCl pH 7.4, 50 mM EDTA), and stretched along superfrost microscope slides, allowing DNA to spread. Slides were air-dried, fixed in methanol and acetic acid (3:1) for 5 min, rehydrated in PBS for 10 min, and denatured with 2.5 M HCl for 30 min at room temperature. Slides were rinsed in PBS and blocked in PBS/0.1%Triton-X100/ 5% BSA for 1 h at room temperature. Slides were then stained using primary rat anti-BrdU (1:100, BU1/75 (ICR1), Abcam, ab6326) and mouse anti-BrdU (1:100 B44, Beckton Dickinson, 347580) antibodies for 1.5 h and then with corresponding secondary goat anti-rat Alexa594 and anti-mouse Alexa488 (A-11007 and A-11001, Thermo Fisher Scientific) antibodies (1:300) for 45 min. Slides were then mounted in Prolong Gold Antifade reagent (Invitrogen) and left to dry overnight. DNA fibers were analyzed using a Carl Zeiss AxioImager Z2 equipped with a CoolCube 1 camera and a ×40/0.75 objective. All images were processed with the ISIS fluorescence imaging system, and DNA fibers were measured using ImageJ. Statistical analyses were carried out using GraphPad Prism and Mann–Whitney non-parametric test.

## HTGTS

High-throughput genome-wide translocation sequencing (HTGTS) was carried out as previously described[32,35,46,73]. Briefly, DNA from activated B cells (day 4 with IL-4, anti-CD40) was sonicated to ~1000 bp (Diagenode Bioruptor) before amplification via an Sμ-specific biotinylated

primer (5'/5BiosG/CAGACCTGGGAATGTATGGT-3'). The biotinylated products were isolated with magnetic beads, ligated to an adaptor, and amplified with a nesting primer 5'-CACACAAAGACTCTGGACCTC-3'. Endonuclease AflII was used before the nesting PCR to remove the germline sequence. The libraries were sequenced on an Illumina Miseq (150 × 150 pair-ended platform). The published HTGTS pipeline was used for mapping and filtering[75]. The best-path searching algorithm related to YAHA[76] was deployed to identify optimal sequence alignments from Bowtie2-reported top alignments (alignment score >50). Mispriming events, germline (unmodified) sequence, sequential joints, and duplicated reads were removed. Duplication was defined by bait, and prey alignment coordinates both within 2 nt of another read's bait and prey alignments. Reads unequivocally mapped to an individual S-region were recovered based on the mappability filter[75]. MHs are defined as sequences that can be assigned to both bait and prey. Insertions are regions that map to neither bait nor prey. The data were then plotted in Excel using a Visual Basic tool[35,46,73].

## Statistics and reproducibility
Biologically independent mice and primary B cells have been used for all experiments as indicated in the figure legends. The number of biologically independent samples has been indicated in the figure legends. All relevant experiments have been performed at least twice, with most of them three or more times. Statistical analyses were performed using unpaired two-tailed t test or two-tailed Mann–Whitney test as indicated in the figure legends.

## Reporting summary
Further information on research design is available in the Nature Portfolio Reporting Summary linked to this article.

## Data availability
The data supporting the findings of this study are available within the paper and its supplementary information files. The RNA-sequencing, HTGTS and CRISPR-Cas9 screen data generated in this study have been deposited in NCBI's Gene Expression Omnibus and are accessible through the GEO Series accession numbers GSE212194, GSE212195, GSE212196, and GSE214643. Source Data for the data presented in graphs are provided with this paper. The uncropped gels and western blots are provided in the Supplementary Information. All data are available from the authors upon request. Source data are provided with this paper.

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

## Acknowledgements

We thank Dr. Richard Baer for helping with the manuscript preparation, and Drs. John Asara from Harvard Medical School and Iok I. Chio at the Institute for Cancer Genetics at Columbia University for helpful discussion on the metabolic analyses. We thank Drs. Barry Sleckman and Bo-Ruei Chen for helping with the CRISPR-Cas9 Screen. We thank Dr. Eric Brown for sharing the Atr null and conditional mouse models. We thank Dr. Teresa Fan, Andre Lane, and Richard Higashi for insightful discussion on nucleotide metabolism. We thank the Genomics Core at the Herbert Irving Comprehensive Cancer Center (HICCC) of Columbia University for technical support in the RNA-seq experiments. This work was supported in part by NIH R01CA158073, R01CA215067, NIH R01CA271595, NIH R01CA275184 and R01CA226852 to S.Z. and 1P01CA174653 to S.Z., R.B., and R.R.; NIH R35CA253126 to R.R.; F31 AI157884 to N.E.B.; AI136581, AI162633, CA254403 to B.K.; D.M. was a senior fellow, and S.Z. was a scholar of Leukemia & Lymphoma Society (LLS). This research used the flow cytometry, molecular cytogenetic, transgenic, and genomic sequencing shared resources funded through the NIH/NCI Cancer Center Support Grant P30CA013696 to the HICCC of Columbia University.

## Author contributions

D.M., H.Z., and S.Z. designed and analyzed the experiments and wrote the paper with proofreading from B.J.L.; D.M. performed all of the experiments for the primary submission and contributed data in the revision. H.Z. helped with RNA-seq analyses and performed new experiments during revision, including the western blotting of ATR, the germinal center analyses, and the apoptosis analyses. B.J.L. carried out the CRISPR-Cas9 screen experiments, and Y.W. analyzed the CRISPR-Cas9 screen results. W.J. generated the Atr-KD allele. N.E.B. and B.K. performed the enzymatic measurement of dNTPs. A.H. shared the software and experience in RNA-seq analyses. J.Z. and R.R. helped with the RNA-seq and CRISPR analyses. S.G. managed the mouse colony and provided support for the animal work.

## Competing interests

The authors declare no competing interests.
