## [Peer Review File · Nature Communications]

ATR kinase supports normal proliferation in the early S phase by preventing replication resource exhaustionREVIEWER COMMENTS

Reviewer #1 (Remarks to the Author):

The manuscript "ATR kinase supports normal proliferation in the early S phase by preventing replication resource exhaustion" by Menolfi et al. explores the role of ATR in DNA replication on unperturbed B cells. The authors take advantage of synchronous cell cycle entry of B cell upon cytokine stimulation to knockout ATR in non-dividing cells and observe the effects of the knockout on DNA replication upon stimulation. While the essential function of ATR in cell cycle has been extensively studied before, using ATR inhibitors and cancer cells, the authors of the current manuscript take it one step further, using primary cultures and a genetic approach. The main conclusion of the current study is that ATR is required to suppress excessive origin firing during DNA replication to prevent nucleotide depletion and subsequent DNA damage. This has been documented in cultured cancer cells before, so the main novelty is demonstrating that the same principles apply in primary B cells. The question why nucleotide complementation failed to restore S-phase progression remains unanswered. While the manuscript contains a vast amount of data and most of the conclusions are justified, the conclusions are mainly confirming the results that were previously reported in cancer cells/cell lines. In order to make it more novel, I propose either looking into some unsolved questions related to the effect of ATR inactivation (insufficiency of nucleotide complementation, effects on the nucleotide balance, etc.), or focusing on possible differences between healthy B cells and cancer.

Additional specific concerns/suggestions:

1. ATR being most important in early S-phase was previously proposed by Zou lab (Buisson et al, 2015). Here the authors conclude that ATR is mainly important for early S-phase cells based on the experiment where ATRi was added at different timepoints during cell cycle progression (figure 2f). On this figure the authors compare 2h ATRi incubation with 48h ATR incubation. Given that the cells enter S-phase less than 24h after activation, the 48h sample has been lacking ATR activity through the whole cell cycle, while 2h sample has been without ATR activity for only 2h. Similarly, on figure S3c, the authors compare 24h or 14h ATRi treatment to 2h. ATRi treatment in early S should be limited to 2h as well – my guess is it would not completely stop EdU incorporation.
2. A better experiment is presented on Fig. 3e, where authors look at the ability of the cells to complete the cell cycle by quantifying the G1 phase cells (this could be repeated and statistics could be presented). However, here we can see that at 24h timepoint when the BrdU pulse is performed, much fewer cells in the ATRi samples are actually incorporating BrdU (13.5% vs. 36%), so this must be taken into account when quantifying the percent of BrdU+ G1 cells at 48h – the percent cells labeled with BrdU at 24h cells making it to G1 (and not the percent of all cells ending up BrdU+ in G1) may actually be higher in the samples with continuous 48h ATRi treatment (possible re-activation of CHK1 by DNAPK proposed by Buisson et al., 2015?). Additionally, the percent of BrdU+ cells appears to increase between 24h and 48h, which is difficult to explain given a short 30 min pulse.
3. Metabolome profile on figure 5d is remarkable. It is showing lower concentrations of the majority of nucleotides in ATR-deficient cells, however, ATP, GTP, UTP, CTP, and dGTP appear to have increased with the loss of ATR, but it not discussed at all. These data indicate specific biosynthetic pathways may be regulated by ATR. A recent preprint from Bakkenist lab (Sugitani et al, 2022) showed regulation of RRM2 by ATR in dividing primary T cells, could a similar mechanism explain this phenotype in B cells?
4. Throughout the manuscript, the authors compare their findings in B cells to the studies in yeast, yet they fail to reference some key studies done in mammalian cells – nucleotide complementation has been done with CHK1 depletion by Gottifredi lab (González Besteiro et al, 2019); CDC7 inhibition has been tried in human cells by Helleday lab (Petermann et al. 2010) with CHK1 inhibitors, the role of CDK1 in ATRi-induced effects has been described by Bakkenist lab (Moiseeva et al, 2019).
5. Specifically, unable to fully complement the effect of CHK1 depletion with nucleotides, González Besteiro et al., 2019 concluded that CHK1 depletion creates replication barriers slowing down replication. It would be very interesting to see if similar barriers can be observed in primary cells lacking ATR.

Minor suggestions:

1. Figures 2a and f have BrdU/PI plots in an unusual orientation. They would be easier to view if PI was on the horizontal axis and BrdU on the vertical, as it is the standard in the field, and as the authors show it on figure 3, for example.

Reviewer #2 (Remarks to the Author):

The study by Menolfi et al. entitled "ATR kinase supports normal proliferation in the early S phase by preventing replication resource exhaustion" aims at understanding the role of ATR in the proliferation of unstressed proliferative B-cells. They initially show that the genetic deletion of ATR, or the loss of its kinase activity, impairs clonal expansion in-vivo, but has no effect on G0-arrested naïve B-cells. To better understand the molecular mechanism by which ATR promotes B cells proliferation, they use a model of naïve B cells ex-vivo culture, stimulated by cytokines (mimicking B-cells activation) and used state-of-the-art techniques such as metabolomics and CRISPR-Cas9 screens, to uncover the determinants of ATR-dependent proliferation. The authors show that cells depleted of ATR, show defects in S-phase progression, and propose that this is due to the function of ATR in fine-tuning origin firing to limit dormant origin firing and prevent a shortage of replication factors, and/or dNTPs. The manuscript tackles an appealing aspect of ATR functions in normal proliferation and uses a relevant physiological model of B-cell activation. These observations are potentially interesting, but the novelty is limited by the fact that the role of ATR and CHK1 in controlling origin firing locally and globally, has already been well established in the context of replicative stress. The study suffers a lack of evidence supporting their views and few technical limitations. Therefore, I do not think that this manuscript is suitable for publication in Nature Communication in its current form.

Major points

1- The authors show that CDK1/CDC7 dual inhibition restores the proliferation of ATR deficient cells. This is a critical result that has potential applications in several pathologies. However, the relevance of these findings needs to be confirmed in an in-vivo setting, such as in the context of naïve B cell clonal expansion

2- Linked to major point 1, do patients with Seckel syndrome suffer from a deficit of naïve/activated B-cells? Is it recapitulated in the mouse model of Seckel syndrome?

3- The authors should show representative images of the DNA fibers, and estimate the inter-origin distance, or the percentage of first-labels origins to support their findings on origin firing.

4- Analysis of cell viability by SSC/FSC gating is not accurate and can be misleading. The authors should confirm those results with a viability dye.

5- In Supplementary Figure 2d, Hydroxyurea did not affect S-phase progression? Was it used at a suboptimal concentration? The authors should confirm that HU and the ATM/DANPKcsi inhibitors hit their targets in this cellular model.

6-Figure 2E, the authors should show the expression of ATR by Western-Blot

7- Can the authors provide more information about the screens, e.g. the raw and adjusted p values of the individual gRNAs? Why use an FDR<0.2 and not the more classically used FDR<0.1 or FDR<0.05?

8- The findings from the screens are very interesting but I see a biological bias that the authors did not comment on. How many of these gRNAs totally prevent origin firing? In this situation, they would

confer resistance to ATRi, not because they are ATR targets but because they impede S-phase entry. For example, I would expect to find gRNAs of genes that induce G1-arrest (Mdm2, p21 ubiquitin ligases,...). Comparing gRNA representation before and after the 6days treatment may clarify those points.

Minor points

1- The authors did not mention the gating strategies used in figure 1a before the B220/CD43 and B220/IgM plots. Did they exclude dead cells (e.g. 7AAC- or DAPI- cells)?

2- The statistical analysis is missing in figure 1b.

3-Figure 7A, the legend is missing

4- the authors show that ATR depletion induces DNA damage in quiescent B-cells. This is rather surprising. Can the authors comment?

Reviewer #3 (Remarks to the Author):

In this manuscript, Zha and colleagues examine the requirement and role of ATR in cell proliferation. Using B cells as a model system to study proliferation, the authors conditionally delete ATR or the ATR kinase activity using stage-specific cre-deletor strains. They find that while loss of ATR is essential for early B cell development in the bone marrow, its loss in naïve B cells does not lead to an overt phenotype under homeostatic conditions. However, when naïve ATR-deficient B cells are activated in vitro, there is a severe defect in B cell proliferation. The authors then carry out a series of PI staining and BrdU-incorporation experiments to demonstrate that the proliferation defect is likely due to massive stalling of replication forks. They further demonstrate that ATR-deficient B cells can initiate DNA replication in early S but they fail to maintain the replicative state by mid-S phase due to reduced nucleotide levels and fork stalling. Finally, the authors show that productive DNA replication can be restored by inhibiting enzymes that suppress origin firing such as CDC7 and CDK1. Taken together, the authors suggest that in unstressed cells, ATR supports proliferation by modulating origin firing in early S-phase cells to avoid exhaustion of nucleotides during the later stages of S phase.

The role of ATR in the cellular DNA damage response following induced DNA damage has been extensively studied for the past several years. It is now clear that replication stress activates ATR and ATR-dependent phosphorylation elicits a concerted response that includes suppression of firing of replication origins. However, the potential role of ATR in regulating origin-firing in unstressed cells have not been fully investigated. In this regard, this study is a detailed characterization of B cells lacking ATR protein or without ATR-kinase activity. The experiments are well done and beyond the cell cycle/proliferation experiments that directly address the ATR-origin firing link, the data from the metabolomic, transcriptomic and the CRISPR/Cas9 screen will be an excellent resource for the field in general. However, the authors should address the issues below to strengthen their study.

1. Fig. 1. and S1: The authors have characterized CSR and the nature of the switch junctions in control and ATR-deleted B cells. The authors should test if the few cells that have switched have done so despite loss of ATR or if the switched cells have escaped cre-mediated deletion.

2. It would be informative if the authors could provide a comprehensive analysis of the B cell phenotype, including germinal center frequency at homeostasis (in Peyer's patches) and following immunization (with say NP-CGG), and the levels of serum Igs.

3. Fig. 2f. The authors claim that ATRi treatment at 46h post-stimulation has no effect on BrdU incorporation. However, the analysis is done 2hrs after ATRi treatment (at 48hs). How much difference is expected in this 2hr window in any case? It is very difficult to interpret this result without data showing how much BrdU normally gets incorporated between 46h and 48hrs.

4. The authors claim that even at 48h post-stimulation, ATR-deficient cells are viable. The authors should include some apoptotic markers to ensure that the cells have not primed to die even though they appear viable.

5. It would be informative if the authors could examine the nature of chromosomal abnormalities in metaphases of ATR-deficient cells at 48h post-stimulation.

6. The authors should be a little conservative in stating that they are testing the role of ATR in unstressed cells. The stimulated B cells are dividing rapidly and have started to express AID which has the potential to cause both Ig and genome-wide DNA breaks. Thus, to refer to ex vivo stimulated B cells as unstressed does not give a true picture of the cell type they are using.

Minor points.

1. The authors should describe the two ATR alleles in more details. They have been described earlier but without any description it was initially difficult to follow the text

2. In Fig. 1b, the authors should provide absolute cell numbers

We thank all the reviewers for their encouragement and their thoughtful suggestions. The reviewers praised the importance and significance of studying normal ATR function in *primary cells* and the value of the definitive genetic approaches used to study an essential gene in our manuscript. The reviewers also pointed out some limitations of our study and made thoughtful and constructive suggestions. In the revised manuscript, we have included **16** new or updated figure panels and tables. They are **Fig. 1c, 1f, 1g, 3f; Sup. Fig. S1d, S1e, S1f, S1j, S2a, S2b, S2c, S2d, S2f, S2h, and S3d, and Sup. Table 1**. We also made all the suggested text modifications and included additional references as suggested by the reviewers. We extended the discussion of our results in the context of prior studies using small chemical inhibitors and transformed human cancer cell lines. We believe that we have addressed all the major concerns of the reviewers. We sincerely appreciate the help and advice from the reviewers and the editors, which significantly improved our manuscript. Enclosed are the point-by-point responses to the review. The original reviewer comments are *italicized and underlined* here for easy identification. New and updated figures are also inserted into the point-by-point response (**with yellow highlight**) for convenience.

REVIEWER COMMENTS

Reviewer #1 (Remarks to the Author):

The manuscript "ATR kinase supports normal proliferation in the early S phase by preventing replication resource exhaustion" by Menolfi et al. explores the role of ATR in DNA replication on unperturbed B cells. The authors take advantage of synchronous cell cycle entry of B cell upon cytokine stimulation to knockout ATR in non-dividing cells and observe the effects of the knockout on DNA replication upon stimulation. While the essential function of ATR in cell cycle has been extensively studied before, using ATR inhibitors and cancer cells, the authors of the current manuscript take it one step further, using primary cultures and a genetic approach. The main conclusion of the current study is that ATR is required to suppress excessive origin firing during DNA replication to prevent nucleotide depletion and subsequent DNA damage. This has been documented in cultured cancer cells before, so the main novelty is demonstrating that the same principles apply in primary B cells. The question why nucleotide complementation failed to restore S-phase progression remains unanswered. While the manuscript contains a vast amount of data and most of the conclusions are justified, the conclusions are mainly confirming the results that were previously reported in cancer cells/cell lines. In order to make it more novel, I propose either looking into some unsolved questions related to the effect of ATR inactivation (insufficiency of nucleotide complementation, effects on the nucleotide balance, etc.) or focusing on possible differences between healthy B cells and cancer.

We thank the reviewer for his/her encouragement and for highlighting the value of our study of primary cells and genetic systems. During the revision, we generated v-abl kinase transformed B cell lines carrying the same *Atr^{+C}* and *Atr^{C/-}* alleles (Brown and Baltimore 2003; Ruzankina et al. 2007) and the tamoxifen (4OHT) inducible ER-Cre recombinase to determine the impact of ATR loss in transformed (in contrast to primary) B cells. Upon ATR deletion, there is a moderate loss of viability seen on Day 7 and a moderate accumulation of BrdU-negative S phase cells in the v-abl kinase transformed B cell lines (**New Figure S1j**). But overall, the loss of viability is significantly less than in primary B cells (from 85.8 to 81.8% in 4 days in transformed B cells vs. from 82.1% to 38.7% in primary cells, **Figure S1c**). This delay in cell death is not due to incomplete deletion of ATR, as confirmed by sensitivity PCR assay (**New Figure S1k**). These results highlight the crucial role of ATR kinase in primary cells that initiates the cell cycle from the G0 phase with low metabolic activity and limited reserve. In contrast, transformed cancer cell lines in continuous culture can benefit from resources diluted from prior cell cycles. Consistent with this model, when activated primary B cells were treated with ATR inhibitor at 24hr after activation (instead of

New Figure S1j and S1k shows that v-abl kinase transformed B cell lines can tolerate ATR deletion (verified by PCR in k) much better than primary B cells (in Figure 2). We noted that the null allele (-) is at different exons from the conditional and del allele that is generated from *LoxP* recombination from the conditional allele. Therefore, the null allele appears like WT in the PCR designed to detect the C and del allele (in k).

from the beginning), the disruption to DNA replication (measured by BrdU pulse chase) was less pronounced. See the updated Figure 3e red labeled, ATRi after 24hr vs. the blue labeled ATRi from time zero). We now include this in the text and discussion. We sincerely thank the reviewer for his/her insightful suggestions.

The question about the failure of nucleotide complementation is indeed exciting and challenging at the same time.

Consistent with prior findings in cell lines (Gonzalez Besteiro et al. 2019) and in yeast models (Forey et al. 2020), we were also NOT able to rescue the productive replication in ATR deficient cells with nucleoside supplementation alone. In addition to reporting and confirming this negative finding, we made three steps forward toward the mechanism. **First**, in addition to ribonucleoside (N) used in prior studies, we supplemented the cells with deoxy-nucleotide (dN) that bypasses the Ribonucleotide reductase (RNR). This approach is uniquely suitable for lymphocytes since lymphocytes have high expression and activity of deoxycytidine kinase (dCK) that can directly phosphorylate deoxynucleosides (dNs) to mono-phosphate nucleotides (dNMPs), while most other cell types can only phosphorylate nucleosides (ribo form) and then depend on RNR to convert NDP to dNDP (see diagram on the right). Indeed, mouse models lacking dCK display specific lymphocytopenia without affecting other major tissues (Toy et al. 2010). Still, dN cannot restore productive replication in ATR-deficient B cells, suggesting RNR is not the only rate-limiting factor. **Second**, with help from Dr. Baek Kim, we measured the intracellular dNTP levels (Figure 6a) before and after dN supplementation and demonstrated that the dN supplementation successfully restored the dNTP levels in ATR-deficient cells. This has not been done in the previous studies and has left intake or conversion defects as possible reasons for the rescue failure. In this regard, we also showed that the dN supplementation significantly attenuated the DNA damage response (Figure 6c – phosphorylation of Kap1, H2AX, and RPA) and DNA strand breaks (Figure 6d - comet assay) in *Atr*-deficient primary B cells, further supporting the impact. **Third and perhaps most importantly**, we found through CRISPR screen and validated with small molecule inhibitors that partially suppressing origin firing successfully rescues productive DNA replication in *Atr*-deficient cells (Figure 6e-g and Figure 7), measured by efficient BrdU incorporation, reduced DNA damage responses (Figure 7d), and most strikingly, **the normalization of dNTP levels in the cells** (Figure 7g). To our knowledge, this is the first time the restoration of the dNTP levels in ATR-deficient cells has been documented along with productive replication. Collectively, these results support a new model in which the overuse of dNTP and the lack of other replication factors, including but not limited to dNTPs, causes replication failure in the ATR-deficient cells. This model explains why nucleotide supplementation has failed to rescue DNA replication without ATR. We thank the reviewer for his/her thoughtful suggestions, and have now added a paragraph to discuss this in the text.

The diagram shows the simplified pathways from ribonucleoside (N) or deoxyribonucleoside (dN) to the dNTP. Lymphocytes have very active dCK that can effectively convert dN to dNMP, therefore bypassing RNR that was down regulated in ATR-deficient cells.

The diagram shows the simplified pathways from ribonucleoside (N) or deoxyribonucleoside (dN) to the dNTP. Lymphocytes have very active dCK that can effectively convert dN to dNMP, therefore bypassing RNR that was down regulated in ATR-deficient cells.

Additional specific concerns/suggestions:

1. ATR being most important in early S-phase was previously proposed by Zou lab (Buisson et al, 2015). Here the authors conclude that ATR is mainly important for early S-phase cells based on the experiment where ATRi was added at different timepoints during cell cycle progression (figure 2f). On this figure the authors compare 2h ATRi incubation with 48h ATR incubation. Given that the cells enter S-phase less than 24h after activation, the 48h sample has been lacking ATR activity through the whole cell cycle, while 2h sample has been without ATR activity for only 2h. Similarly, on figure S3c, the authors compare 24h or 14h ATRi treatment to 2h. ATRi treatment in early S should be limited to 2h as well – my guess is it would not completely stop EdU incorporation.

Indeed, we have thought about and tried these experiments suggested by the reviewer. The results are now included in the updated Figure 3e. Briefly, we stimulated the primary B cells and split the culture into four groups and added ATRi at 24hr after initial stimulation for ~1 hr (green line and green label) or for 24hr (red

line and red label) and chased the BrdU+ cells (pulse-labeled for 30 min) for the next 24hr. As expected from the ATRi supplemented at 22h, the one hour treated and chase has no measurable impact on S phase progression. In contrast, when ATRi was added at the beginning (blue line), there was a significant block in the S phase progression. We also repeated the experiments independently a few times. **The data is now included in the updated Figure 3e and quantified in New Figure S3d.**

Updated Figure 3e and S3d: **3e)** Primary B cells were stimulated together and split in four different cultures. When ATRi was added at 24h post-stimulation, either for 1h (green treatment) or for the next 24h until cell collection at 48h (red treatment), BrdU incorporation and recovery resemble the one of the untreated ctrl. Comparable BrdU-positive G1 sub-populations were observed within the total of S phase cells, suggesting no defects in cell cycle progression and re-entry. However, when ATRi was added in G0, at the beginning of the experiment (blue treatment), cell cycle re-entry was severely compromised, revealing the essential function of ATR. The quantification of this data is now included in Figure 3e (shown as percentage of G1 cells among the BrdU+ cells as suggested by Reviewer 1). **S3d)** As suggested by the reviewer, we quantified the % of G1 BrdU+ cells versus total BrdU-labelled positive cells. CD21-Cre AtrC/- and ctrl cells treated with ATRi for the length of the experiment display a significant reduction of G1 BrdU+ cells, indicative of defects in cell cycle re-entry. Two or three independent experiments are quantified, and the graph refers to the representative flow cytometry analyses reported in Figure 3e. The quantification is reported in Supplementary Figure 3d.

2. A better experiment is presented on Fig. 3e, where authors look at the ability of the cells to complete the cell cycle by quantifying the G1 phase cells (this could be repeated and statistics could be presented). However, here we can see that at 24h timepoint when the BrdU pulse is performed, much fewer cells in the ATRi samples are actually incorporating BrdU (13.5% vs. 36%), so this must be taken into account when quantifying the percent of BrdU+ G1 cells at 48h – the percent cells labeled with BrdU at 24h cells making it to G1 (and not the percent of all cells ending up BrdU+ in G1) may actually be higher in the samples with continuous 48h ATRi treatment (possible re-activation of CHK1 by DNAPK proposed by Buisson et al., 2015?). Additionally, the percent of BrdU+ cells appears to increase between 24h and 48h, which is difficult to explain given a short 30 min pulse.

We thank the reviewer for his/her thoughtful suggestion. We did have independent repeats and now also quantified them as a percentage of BrdU+ cells (control for BrdU labeling percentage difference). The results are now provided in updated Figure S3d. We also note that while ATRi treatment from time zero reduced BrdU+ cell percentage (potentially due the dominant negative role of the inactive ATR protein previously reported (Menolfi et al. 2018)), Atr KO B cells (CD21Cre+AtrC/-) had the same percentage of BrdU+ cells at 24h as the control AtrC/+ B cells (Updated Figure 3e and S3d), while the G1% among BrdU+ cells markedly reduced in Atr-KO cells. We thank the reviewer for noting the increased BrdU+% during chasing in some experiments. We only saw this in primary B cells and have tried multiple experimental conditions, which yielded variable results. One possibility might be the BrdU negative population that reflects non-responders to cytokine stimulation might have dropped out during the chases. As seen in Figure S1C, by day 4 the viability for AtrC/+ cell is only 75% and for AtrC/- cells is ~40%.

We also noted the results from the Buisson et al., 2015 paper. Indeed, we have measured pCHK1 at both 24hr (Figure S3g) and 48hr (Fig. 2E) after activation. In neither case did we observe significant phosphorylation of CHK1. We noted that Buisson et al. performed their experiments in human cancer cell lines (Buisson et al. 2015). Here all our experiments were carried out in primary murine cells. We and others have previously shown that the protein levels and activities of DNA-PKcs and KU (the DNA binding component of DNA-PK) are

50-100 fold lower in mouse cells than in human cells (Jiang et al. 2019). This might explain the lack of DNA-PK-mediated CHK1 phosphorylation in our murine-based experiments. We now mention this in the text. We are very interested in this species-specific difference but think that this is beyond the scope of the current manuscript.

3. Metabolome profile on figure 5d is remarkable. It is showing lower concentrations of the majority of nucleotides in ATR-deficient cells, however, ATP, GTP, UTP, CTP, and dGTP appear to have increased with the loss of ATR, but it not discussed at all. These data indicate specific biosynthetic pathways may be regulated by ATR. A recent preprint from Bakkenist lab (Sugitani et al., 2022) showed regulation of RRM2 by ATR in dividing primary T cells; could a similar mechanism explain this phenotype in B cells?

We thank the reviewer for pointing out the increase in NTPs. We also noted it in the main text. While we do not know the exact cause of the moderate NTP increase (1.2-1.5 fold), it would be consistent with the lack of rescue by nucleosides (ribo form) reported previously (Gonzalez Besteiro et al. 2019). It is possible that the lack of NTP usage in some pathways contributed to this accumulation. Alternatively, there might be excess conversion to NTP from NMP and NDP. As reported by Bakkenist and colleagues (Sugitani et al. 2022), we also observed a 1.2-1.5 fold reduction of the RRM1 and RRM2 mRNA levels in the *Atr*-deficient B cells (Figure S5e). This and other observations encouraged us to supply the cells with deoxynucleosides (dN) rather than nucleosides to bypass RNR (see the response to reviewer 1 general comments and diagram) (Lane and Fan 2015). While dN successfully increases deoxy nucleotide levels in the cells, it could not rescue productive DNA replication (measured by BrdU incorporation) (Figure 6).

4. Throughout the manuscript, the authors compare their findings in B cells to the studies in yeast, yet they fail to reference some key studies done in mammalian cells – nucleotide complementation has been done with CHK1 depletion by Gottifredi lab (González Besteiro et al, 2019); CDC7 inhibition has been tried in human cells by Helleday lab (Petermann et al. 2010) with CHK1 inhibitors, the role of CDK1 in ATRi-induced effects has been described by Bakkenist lab (Moiseeva et al., 2019).

We thank the reviewer for pointing out those key references that we missed. We have now cited them all in the main text- *González Besteiro et al., 2019- Ref 38, Petermann et al., 2010 -Ref 61; and Moiseeva et al., 2019- Ref 62.* We further note that CHK1 and ATR have independent functions beyond their well-recognized epistatic roles. In contrast to CHK1, which is transcriptionally silenced and absent in G0 quiescent cells (Figure S3g), ATR is transcribed in G0 arrested primary cells (Lee et al. 2014a). Moreover, the three references all use cancer cell lines treated with either small molecule inhibitors for ATR or CHK1, highlighting the knowledge gap in non-transformed and genetically deleted cells.

Together with prior findings, our data collected on primary B cells show that the ATR kinase activity suppressing the excessive firing of the replication origins to coordinate S phase progression is a conserved and essential function of ATR in both normal and transformed cells.

5. Specifically, unable to fully complement the effect of CHK1 depletion with nucleotides. González Besteiro et al., 2019 concluded that CHK1 depletion creates replication barriers slowing down replication. It would be very interesting to see if similar barriers can be observed in primary cells lacking ATR.

González et al. defined "the barrier" in part by replication fork asymmetry (Gonzalez Besteiro et al. 2019). Following the reviewer's suggestion, we analyzed fork asymmetry in ctrl and *Atr*-deficient primary B cells (New Figure S2d). DNA fiber experiments were performed in 24h or 48h activated B cells, labeled for 30 min with CldU and 30' with IdU. Only replication tracks where two IdU fibers (green) flanking the same CldU track (red) were measured. We calculated the ratio

Updated Figure S2d DNA replication fork symmetry was calculated for CD21-Cre Atr+/C and AtrC/- B cells. Representative images are shown and quantification of the ratio between two IdU forks originating from the same CldU track is shown. Unpaired two tailed student's t-test was used for the statistical analysis.

between each paired IdU fiber and considered ratio =1 as a symmetric fork and ratio >1 as fork asymmetry. The ratio was computed using the longer IdU fiber as the numerator and the shorter as the denominator. The data indicate that similar to what has been previously reported for CHK1 inhibitor-treated cells, replication forks are asymmetric in *CD21-Cre⁺Atr^{C/-}* cells compared to ctrl cells. We now report this in new Figure S2d.

Minor suggestions:

1. Figures 2a and f have BrdU/PI plots in an unusual orientation. They would be easier to view if PI was on the horizontal axis and BrdU on the vertical, as it is the standard in the field, and as the authors show it on figure 3, for example.

We have re-plotted the BrdU/PI plots (Updated Figure 2a and 2f, as well as Figure S2a and S2b) to make them vertical and consistent with the standard view in the field in all main and supplementary figures.

Updated Figure 2a and 2f show accumulation of BrdU negative cells with S phase DNA content in Atr-deficient (2a) or ATR inhibitor treated (2f) primary B cells.

Reviewer #2 (Remarks to the Author):

The study by Menolfi et al. entitled "ATR kinase supports normal proliferation in the early S phase by preventing replication resource exhaustion" aims at understanding the role of ATR in the proliferation of unstressed proliferative B-cells. They initially show that the genetic deletion of ATR, or the loss of its kinase activity, impairs clonal expansion in-vivo, but has no effect on G0-arrested naïve B-cells. To better understand the molecular mechanism by which ATR promotes B cells proliferation, they use a model of naïve B cells ex-vivo culture, stimulated by cytokines (mimicking B-cells activation) and used state-of-the-art techniques such as metabolomics and CRISPR-Cas9 screens, to uncover the determinants of ATR-dependent proliferation. The authors show that cells depleted of ATR, show defects in S-phase progression, and propose that this is due to the function of ATR in fine-tuning origin firing to limit dormant origin firing and prevent a shortage of replication factors, and/or dNTPs. The manuscript tackles an appealing aspect of ATR functions in normal proliferation and uses a relevant physiological model of B-cell activation. These observations are potentially interesting, but the novelty is limited by the fact that the role of ATR and CHK1 in controlling origin firing locally and globally, has already been well established in the context of replicative stress. The study suffers a lack of evidence supporting their views and few technical limitations. Therefore, I do not think that this manuscript is suitable for publication in Nature Communication in its current form.

We thank the reviewer for carefully elevating our manuscript and for his/her candid feedback. We note that ATR (and CHK1 and other proteins) has many functions during replication stress. But not all of these functions are essential for normal unstressed primary B cells. While it is important to understand the role of ATR under replication stress or in cancer cells, the success of ATR inhibition as a therapeutic approach also requires a thoughtful evaluation of ATR function in normal unstressed tissues and cells *in vivo*. To some degree, it is satisfying and comforting to know that primary cells did NOT re-invent the wheel. Instead, the conserved function of ATR in suppressing excessive origin firing (among many other functions of ATR) underlies the essential role of ATR in proliferation. With this knowledge, it would be possible to target other cancer-specific and potentially replication stress-specific functions of ATR for cancer therapy while sparing normal tissue/cells.

Major points

1- The authors show that CDK1/CDC7 dual inhibition restores the proliferation of ATR-deficient cells. This is a critical result that has potential applications in several pathologies. However, the relevance of these findings needs to be confirmed in an in-vivo setting, such as in the context of naïve B cell clonal expansion

We agree with the reviewer. To determine the *in vivo* implication of *Atr*-deficiency in B clonal expansion, we adopted a well-established immunization model (Klein et al. 2003) to examine the impact of *Atr*-deficiency in naïve B cell germinal center response after antigen exposure (e.g., sheep red blood cells- SRBC). In *CD21^{Cre}Atr^{C/-}* mice, both spontaneous and induced germinal center B cells (GL7+ and CD95+) – the fraction of B cells activated by T cells upon antigen exposure *in vivo* - were significantly reduced (New Figure 1f and 1g), supporting a critical role of ATR in B cell clonal expansion. While there are many clinical-grade CDK inhibitors, most target CDK4/6 or CDK12/13. Unfortunately, while the current CDC7 inhibitor we used is in a phase I clinical trial, the CDK1 inhibitor is unsuitable for *in vivo* application, preventing us from rescuing the germinal center response *in vivo*.

New Figure 1f (left) and 1g (right - bar graph) show the frequency of activated germinal center B cells (marked by GL7+ and CD95+) among all splenic B cells has significantly reduced both at the basal line (naïve) and upon immunization (sheep red blood cells). The IHC staining for the germinal center marker Bcl6 validated the flow cytometry findings. On the right, the number of germinal center B cells after immunization was plotted from 4 independent mice of each genotype (average and standard errors were plotted). The student t-test was used to determine the p-value.

2- Linked to major point 1, do patients with Seckel syndrome suffer from a deficit of naïve/activated B-cells? Is it recapitulated in the mouse model of Seckel syndrome?

We do not have direct access to Seckel syndrome patients. On the OMIM database maintained by NCBI, there are 5 ATR mutation alleles from 4 different case reports of Seckel syndrome patients. They all have variable levels of ATR, consistent with hypomorphic ATR deficiency (<https://omim.org/entry/601215#allelicVariants>). All the patients display dwarfism and microcephaly. The blood cell counts, and more specifically lymphocyte counts, were not discussed in any of the case reports. However, in one study, an EBV-transformed lymphoblastic cell line was derived from a patient carrying compound ATR-heterozygous mutations. The authors comment in the text that the cell line grows slower (Mokrani-Benhelli et al. 2013), consistent with the role of ATR in B cell proliferation.

Meanwhile, the hypomorphic Seckel syndrome mouse model (Murga et al. 2009) does show reduced white blood cell counts (Figure 3e). Moreover, using bone marrow transplantation, the authors showed that *Atr^{S/S}* bone has a significant yet reduced ability to reconstitute the B and T cell lineage specifically (in their Sup. Figure S6e). We note that the authors suggested that hematopoietic niche defects might contribute to the pancytopenia in the Seckel mouse models, further highlighting the value of using lineage-specific and developmental-stage specific Cre alleles to dissect the cell-autonomous and development stage-specific role of ATR in B lymphocytes presented in our study.

Figure 3e - germline mice

Sup. Figure S6e - bone marrow transplantation

Figures from the Seckel cell mouse model (ref 14) documented reduced white blood cell counts in germline mice (Figure 3e in original paper) and reduced ability to reconstitute B and T cells upon transplantation (Figure S6e).

Finally, we note that lymphocytes are not essential for viability or embryonic development. Given the pleiotropic role of ATR in nearly all cell types, it is not surprising that lymphocytes are not the primary cause of morbidity in ATR-mutated patients. This is different in adult animals, where most tissues have stopped proliferation and are insensitive to ATR loss. At the same time, lymphocytes, upon activation, undergo rapid expansion, highlighting the importance and the unique value of B cells-specific conditional ATR deficient alleles. Moreover, we note that B cell development requires rapidly expanding large Pre-B cells after successful IgH heavy chain rearrangement. If ATR is essential for the proliferation of primary cells, the reduced B cell numbers in the Seckel Syndrome model are likely due to development defects (not necessarily naïve/activated B cell defects). The activation defects are crucial for adult patients exposed to ATR inhibition as part of cancer therapy. Here we conditionally deleted ATR only in naïve/mature B cells (after the expansion of pre-B cells), bypassing the early need for ATR in B cell development, and allowing us to directly access the response to ATR loss in naïve primary B cells.

3- The authors should show representative images of the DNA fibers, and estimate the inter-origin distance, or the percentage of "first-labels origins" to support their findings on origin firing.

As suggested by the reviewer, we have included examples of DNA fibers in **New Supplementary Figure 2c and 2d**. Unfortunately, we cannot access the instrument to perform the inter-origin distance measurement in our lab right now. Given our focus on primary B cells freshly isolated from the animal and activated within 24hrs, it would also be challenging to ship the cells to outside collaborators for this experiment. We have discussed these caveats in our paper and cited others' publications in which inter-origin distance has been measured in the context of ATR inhibition in murine and human cells (Moiseeva et al. 2019; Sugitani et al. 2022).

4- Analysis of cell viability by SSC/FSC gating is not accurate and can be misleading. The authors should confirm those results with a viability dye.

Following this reviewer's comment, we compare the SSC/FSC gated live population with the standard PI-Annexin staining in murine B lymphocytes. As seen in this figure for review only, in murine B lymphocytes, parallel analyses for SSC/FSC and PI-Annexin show that SSC/FSC has a ~98% accuracy in detecting apoptotic cells (the sum of early and late apoptosis). 98.9% or 98.4% of dead cells (determined by SSC/FSC combination) are stained positive for PI or Annexin. Meanwhile, 94.7% and 80% of live cells (measured by SSC/FSC) are negative for PI and Annexin. Indeed, in the field of immunology, using

Figure for reviewer 2: Primary or Etoposide treated (positive control) B lymphocytes were stained in parallel for PI-Annexin or FSC/SSC. When gated on the dead cells based on FSC/SSC, 98% of the cells are positive for Annexin and/or PI. When gated on the "live" cells based on FSC/SSC, 94.7% or 80% of the cells are indeed negative for both PI and Annexin. The results demonstrate excellent ability for FSC/SSC to separate the live vs dead lymphocytes.

SSC/FSC to gate out live vs. dead cells is routine. This super ability for SSC/FSC to distinguish live vs. death in lymphocytes is partly due to the nearly perfect sphere shape and small cytoplasm of lymphocytes, which give little noise in SSC measurements.

Nevertheless, we activated B cells from three or more independent CD21Cre+Atr+/C and CD21Cre+AtrC/- mice and performed apoptosis analysis using Annexin and PI. The result is presented in **New Figures S1e and 1f** and shows that, indeed the activated ATR-deficient lymphocytes undergo apoptosis.

New Figure S1e and S1f: Representative Flow cytometry for apoptotic markers (PI and Annexin) in activated lymphocytes (S1e) and quantification (S1f) for different days after stimulation. The data proved that the Atr-deficient B cells indeed died of apoptosis and demonstrate excellent ability for FSC/SSC to separate the live vs dead lymphocytes.

5- In Supplementary Figure 2d, Hydroxyurea did not affect S-phase progression? Was it used at a suboptimal concentration? The authors should confirm that HU and the ATM/DANPKcsi inhibitors hit their targets in this cellular model.

We thank the reviewer for this comment. Original Supplementary Figure 2d is now Figure S2g. We indeed used a low dose of HU (20 μM instead of 2mM) to allow the cells to successfully enter S phase. Otherwise, we would not be able to measure BrdU levels in cells with S phase DNA content (the goal of this experiment). We also performed experiments with a high dose of HU 2 mM, which prevented S phase entry as shown in **New Supplementary Figure 2h**. Therefore, HU did work appropriately. We have used commercial (from Selleckman) ATMi, and DNA-PKi at the dose indicated (7.5-15uM)(Crowe et al. 2018; Jiang et al. 2019; Crowe et al. 2020; Shao et al. 2020; Wang et al. 2020) and always confirmed that ATMi reduced Ig class switch

New Figure S2h: High dose HU completely blocked DNA replication. To quantify the frequency of BrdU negative S phase cells, we have to move to a lower dose of HU.

recombination (CSR) by ~50% (Lumsden et al. 2004; Reina-San-Martin et al. 2004; Pan-Hammarstrom et al. 2006) as also reported in the figure for reviewers only. In comparison, DNA-PKcs inhibitor reduced CSR by ~10-20%, consistent with DNA-PKcs null cells (Manis et al. 2002; Crowe et al. 2018; Crowe et al. 2020). Given the amount of data and figures in the current manuscript, we felt that it is not necessary to include these technical validations of commercially available and well-characterized inhibitors in the main data. We and others have also reported that ATM null (Lumsden et al. 2004; Reina-San-Martin et al. 2004; Pan-Hammarstrom et al. 2006) or DNA-PKcs null (Manis et al. 2002; Crowe et al. 2018; Crowe et al. 2020) B cells show no proliferation defects. We have now cited those papers in the revised manuscript.

Figure for reviewers. Upon activation, CSR cells were either left untreated or treated with ATMi (7.5 μM). Cells were collected after 4 days in CSR medium and analyzed by flow cytometry for IgG1 and B220. ATMi treatment led to ~50% reduction in CSR efficiency.

6-Figure 2E, the authors should show the expression of ATR by Western-Blot.

Unfortunately, there is no reliable antibody for mouse ATR. The original antibody used by Dr. Eric Brown in his first publication on ATR KO mice is not available anymore. Despite several efforts, including reaching out and testing older stocks provided by Dr. Brown's lab, we failed to detect mouse ATR. Meanwhile, we note that this ATR conditional allele has been very well characterized since 2000 (Brown and Baltimore 2000; Brown and Baltimore 2003; Ruzankina et al. 2009; Onksen et al. 2011), and we have documented robust deletion in primary B cells via PCR (Figure S1b and S1k). Perhaps most importantly, deletion of ATR in pre-B cells via Mb1Cre completely abrogated B cell development *in vivo*, and deletion of ATR in naïve B cells via CD21Cre blocked B cell activation within one cell cycle and erased CHK1 phosphorylation (Figure 1 and Figure 2). Altogether, these solid and undeniable phenotypes, together with a sensitive PCR assay for ATR deletion, indicate that ATR is indeed deleted in naïve B cells. We noted that naïve B cells could stay in the peripheral organs for months, if not years, which would be more than sufficient to overcome the long half-life of ATR proteins.

7- Can the authors provide more information about the screens, e.g. the raw and adjusted p values of the individual gRNAs? Why use an FDR<0.2 and not the more classically used FDR<0.1 or FDR<0.05?

The raw data and the adjusted p-values for consistent hits has been included in **Supplementary table 1**, now with updated highlights. Moreover, the entire CRISPR dataset has been made available at GSE214643. We opted to use FDR< 0.2 because, 1) several recent CRISPR Screens published on DNA repair and genomic instability use this standard (Hart et al. 2017; Noordermeer et al. 2018; Zimmermann et al. 2018; Olivieri et al. 2020) and 2) the developer of the MAGECK pipeline (<https://sourceforge.net/p/mageck/wiki/Home/>) that we used to analyze the CRISPR/Cas9 screen suggests this FDR for identifying **recurrent** targets in multiple independent screens (3 ATR inhibitor and 1 CHK1 inhibitor screen here). The detailed data including adjusted FDR, Z and beta scores of the hits are included in **updated Sup. Table 1**.

8- The findings from the screens are very interesting but I see a biological bias that the authors did not comment on. How many of these gRNAs totally prevent origin firing? In this situation, they would confer resistance to ATRi, not because they are ATR targets but because they impede S-phase entry. For example, I would expect to find gRNAs of genes that induce G1-arrest (Mdm2, p21 ubiquitin ligases,...).

We thank the reviewer for this comment. We do see a few genes associated with G1 cell cycles (e.g., Cyclin D3 in individual screens). We do not expect to see targets that "completely" block origin firing. CRISPR screening is an enrichment screen based on cell proliferation ability. If the cells are completely arrested or die, we would not see any enrichment. Similarly, our rescue with CDC7i and CDK1i used a sub-lethal dose. A high dose of CDC7i and CDK1i would completely block WT B cells. In fact, we chose the dose of CDC7 and CDK1 inhibitors that do NOT affect the proliferation of control B cells. Finally, we noted that primary B cells have a different p53 response than common epithelial cells. Specifically, upon radiation-induced p53 activation, primary lymphocytes activate apoptosis rather than cell cycle arrest. For this reason we also did not observe p21 itself as a reliable p53 target in primary lymphocytes.

Minor points

1- The authors did not mention the gating strategies used in figure 1a before the B220/CD43 and B220/IgM plots. Did they exclude dead cells (e.g. 7AAC- or DAPI- cells)?

The dead cells were excluded using FSC/SSC, and red blood cells were excluded using a TER119 antibody. We now discussed this in the methods. As shown in response to reviewer 2 major comment 4, in lymphocytes, SSC and FSC are reasonably good indicators for viability. That information is now included in the online method section.

2- The statistical analysis is missing in figure 1b.

We now included the statistical analysis for B cell populations and also absolute bone marrow cell counts (updated Figure 1b and 1c).

Updated Figure 1b and 1c : 1b(left)

Mb1+/Cre deletion leads to a significant loss of immature, recirculating and naïve B cells in Atr-deficient mice (AtrC/- and AtrC/KD). Statistical analysis was done using unpaired two tail t test. On the contrary, CD21-Cre induced deletion does not lead to any B cell development defects. The t tests between any of the pairs tested were not significant. 1c (right); the absolute bone marrow cell counts from CD21cre AtrC/- and control mice. There is no significant difference across all populations.

3-Figure 7A, the legend is missing

Figure 7A legend has been expanded to include more details now.

4- the authors show that ATR depletion induces DNA damage in quiescent B-cells. This is rather surprising. Can the authors comment?

Indeed, we were very interested in this observation. Literature suggests that ATR might participate in nucleotide excision repair(Lee et al. 2014b) and be activated by transcription-generated single-strand RNA(Gorthi et al. 2018). In Figure S3h, we showed that despite the increased alkaline comet signals, we did not find significant phosphorylation of KAP1 or RPA, likely consistent with low levels of single-strand nicks. We feel that this discussion is beyond the scope of this already very complicated paper, but interested in pursuing this question in future studies.

Reviewer #3 (Remarks to the Author):

In this manuscript, Zha and colleagues examine the requirement and role of ATR in cell proliferation. Using B cells as a model system to study proliferation, the authors conditionally delete ATR or the ATR kinase activity using stage-specific cre-deletor strains. They find that while loss of ATR is essential for early B cell development in the bone marrow, its loss in naïve B cells does not lead to an overt phenotype under homeostatic conditions. However, when naïve ATR-deficient B cells are activated in vitro, there is a severe defect in B cell proliferation. The authors then carry out a series of PI staining and BrdU-incorporation experiments to demonstrate that the proliferation defect is likely due to massive stalling of replication forks. They further demonstrate that ATR-deficient B cells can initiate DNA replication in early S but they fail to maintain the replicative state by mid-S phase due to reduced nucleotide levels and fork stalling. Finally, the authors show that productive DNA replication can be restored by inhibiting enzymes that suppress origin firing such as CDC7 and CDK1. Taken together, the authors suggest that in unstressed cells, ATR supports proliferation by modulating origin firing in early S-phase cells to avoid exhaustion of nucleotides during the later stages of S phase.

The role of ATR in the cellular DNA damage response following induced DNA damage has been extensively studied for the past several years. It is now clear that replication stress activates ATR and ATR-dependent phosphorylation elicits a concerted response that includes suppression of firing of replication origins. However, the potential role of ATR in regulating origin-firing in unstressed cells have not been fully investigated. In this regard, this study is a detailed characterization of B cells lacking ATR protein or without ATR-kinase activity. The experiments are well done and beyond the cell cycle/proliferation experiments that directly address the ATR-origin firing link, the data from the metabolomic, transcriptomic and the CRISPR/Cas9 screen will be an

excellent resource for the field in general. However, the authors should address the issues below to strengthen their study.

We thank this reviewer for his/her encouragements. We have addressed the specific comments below.

1. Fig. 1. and S1: The authors have characterized CSR and the nature of the switch junctions in control and ATR-deleted B cells. The authors should test if the few cells that have switched have done so despite the loss of ATR or if the switched cells have escaped cre-mediated deletion.

We thank the reviewer for his/her thoughtful suggestions. Indeed, we performed PCR genotyping on different days after CSR routinely. There is a variable amount of residual ATR conditional allele that often flares up in D3 or D4 only in the *Atr^{C/-}* mice. This is consistent with the strong selection against ATR deleted cells and the accumulation of the escapers. But as seen in the representative figure, the deletion band is always the dominant band in the PCR, suggesting the majority of the cells have deleted ATR. A similar observation has been made by us and others on CtIP, another essential gene in activating B cells (Polato et al. 2014; Liu et al. 2019; Wang et al. 2020). We have now included one representative data here for the review.

PCR Genotyping during activation for the Review.

Here we show in the course of the 4 day stimulation, the conditional bands remain at very low levels if any. The rather prominent aspecific band on D0 might reflect impurity during magnetic beads purification, which usually achieves 90-95% B220+ cells. During the course of stimulation, the deletion is best at D1 and D2, and in the *Atr^{C/-}* B cells, after D4, there are very few viable cells left, rendering the PCR unreliable.

2. It would be informative if the authors could provide a comprehensive analysis of the B cell phenotype, including germinal center frequency at homeostasis (in Peyer's patches) and following immunization (with, say, NP-CGG) and the levels of serum Igs.

Due to the small size of our animal colony after COVID related shutdown (we lost 80% of our mice in 3 days), we have only a limited batch and ctrl mice for these experiments right now. Instead of the antigen-specific immunization (NP-CGG), which usually yields small germinal centers and viable antibody titers, we chose to immunize the mice with sheep-red-blood cells that generate a robust and reliable germinal center response (new Figure 1f and 1g). Indeed, the frequency of GC B cells (CD95+ and GL7+) is reduced in CD21Cre+*Atr^{C/-}* mice. Moreover, upon immunization, the induced GC cell frequency is also lower in CD21Cre+*Atr^{C/-}* mice. Together the data provide strong evidence for the critical role of ATR in germinal center response. Due to the number of animals needed and the duration required for multiple rounds of immunization to measure antigen-specific Ig titers, we were not able to measure antigen-specific serum Ig levels *in vivo*. But we felt that the additional evidence on both Germinal center flow cytometry and histology analyses before and after immunization uniformly point to a B cell maturation defect associated with ATR loss.

3. Fig. 2f. The authors claim that ATRi treatment at 46h post-stimulation has no effect on BrdU incorporation. However, the analysis is done 2hrs after ATRi treatment (at 48hs). How much difference is expected in this 2hr window in any case? It is very difficult to interpret this result without data showing how much BrdU normally gets incorporated between 46h and 48hrs.

When performing BrdU labeling, all the cells that were in the S phase during the labeling window will be BrdU positive. Similarly, when we treated the cells with ATR inhibitor for 2 hrs, all the cells, including all S phase cells, will lose significant ATR activity in this window. The point of this updated Figure 2f is to prove that ATR is not essential for the ongoing S phase and on-going DNA replication; as such, inhibiting ATR does NOT immediately stop BrdU incorporation as high dose HU would do (New Figure S1h). We now better explain this in the text.

We also treated the cells for 24h (from 24hr to 48hr after activation) and chased the BrdU+ cells (S phase cells during the BrdU pulse label). Updated Figure 3e and New Figure S3d (quantification) – see the response to review 1 comment 1 - now show that once the cells have started proliferation, ATR inhibition has less immediate impacts.

4. The authors claim that even at 48h post-stimulation, ATR-deficient cells are viable. The authors should include some apoptotic markers to ensure that the cells have not primed to die even though they appear viable.

We have now performed Annexin-PI staining for the activated B cells (Control and Atr deficient) (shown in new Figure S1e and S1f (see response to reviewer 2 major comment 4)). There is significant cell death, so the analyses focus on the 50% viable cells. We noted that viability measured by FSC/SSC nicely aligns with those measured by Annexin-PI. The viable cells defined by FSC/SSC are 98+% viable using Annexin V and PI as the marker. (see response to reviewer 2 major comment 4).

5. It would be informative if the authors could examine the nature of chromosomal abnormalities in metaphases of ATR-deficient cells at 48h post-stimulation.

We thank the reviewer for his/her suggestion. We have performed Telomere-FISH in CD21-Cre Atr+/C, AtrC/- and AtrC/KD activated B cells at D3 post-stimulation. We chose to use T-FISH to enhance the sensitivity to

New Figure S2f Top panel shows the frequency of chromosomal breaks identified by Telomere FISH. Lower panel shows the level of telomere instability-defined by more than 1 telomere signal per chromosome end.

6. The authors should be a little conservative in stating that they are testing the role of ATR in unstressed cells. The stimulated B cells are dividing rapidly and have started to express AID which has the potential to cause both Ig and genome-wide DNA breaks. Thus, to refer to ex vivo stimulated B cells as unstressed does not give a true picture of the cell type they are using.

We thank the reviewer for this insightful comment. Indeed, AID expression might add another layer of instability. Prompted by this suggestion, we searched for when AID is expressed in activated B cells. We found the following references that indicate that AID protein is not detected in the first 24 hours after activation (Schrader et al. 2005; McBride et al. 2006). We have noted this in the paper and focused most data, including metabolome, RNA-seq, and dNTP data, on the first 24 hours after stimulation.

Minor points.

1. The authors should describe the two ATR alleles in more details. They have been described earlier but without any description it was initially difficult to follow the text

We thank the reviewer for pointing this out. We have now included a detailed discussion about the alleles in the online method section. The Atr KO allele made by Dr. Eric Brown in the Baltimore lab replaced exon 1-3 of Atr with a Neo-R cassette. The ATR conditional allele and the corresponding Del (Δ) allele (after loxP recombination) introduced two LoxP sites flanking the C-terminal exons encoding the kinase domain. Since the Atr^{C/Δ} and Atr^{KO} involve distinct exons of the Atr gene, the PCR genotyping designed for Atr^{C/Δ} would show a “WT” band in Atr^{KO} cells. Similarly, PCR genotyping designed for Atr^{KO} would detect the intact exon 1-3 in the

detect chromosome breaks, since mouse chromosomes have short arms, and loss of chromosome fragments can easily be missed without telomere markers (Franco et al. 2006). ATR-deficient B cells show a small but consistent increase in aberrations per metaphase, in particular, breaks and fragile telomeres. Given the severe cell cycle arrest, the very low level of mitotic cells, and the fact that the chromosomal abnormality was measured in metaphase, we believe that this likely under-estimates the actual number of breaks and instability in the S phase or G2 phase cells. Nevertheless, we have included the data in New Figure S2f and discussed them in the text.

Figure 2B from McBride K et al. PNAS 2006

Figure 1A from Schrader et al. JEM 2005

Atr^{C/A} mice. This can be seen in New Supplementary Figure 1k where the v-abl kinase transformed B cell line 8412 is *Atr*^{C/KO}. In both cases, the ATR protein cannot be detected, resulting in a complete loss of ATR. In our studies, we used the *Atr*^{KO} and *Atr*^A alleles interchangeably and referred to them as *Atr* together for simplicity.

2. In Fig. 1b, the authors should provide absolute cell numbers

We have now included the absolute B cell number from one femur per mice in new Figure 1c for CD21Cre *Atr*^{C/-} and *C/+* mice (see page 9 of the point-by-point response). During COVID, we were asked to euthanize 80% of our colony in 3 days. Since we have collected all the Mb1Cre+*Atr*^{C/-} data before COVID and knew that there are no naïve B cells in the Mb1Cre+*Atr*^{C/-} mice for further study, we have not re-established the experimental Mb1Cre+*Atr*^{C/-} colony. As a result, we currently do not have live animals at the proper age for cell count analyses. But since Mb1 is a B cell-specific Cre (Hobeika et al. 2006) and does not affect T cells and other lineages, the percentage of B cells in bone marrow provides an objective estimation of the B cell defects in those mice. We apologize for the lack of those data. We believe it would not jeopardize the overall conclusion, supported by Figures 1a, 1b.

References

- Brown EJ, Baltimore D. 2000. ATR disruption leads to chromosomal fragmentation and early embryonic lethality. *Genes Dev* 14: 397-402.
- Brown EJ, Baltimore D. 2003. Essential and dispensable roles of ATR in cell cycle arrest and genome maintenance. *Genes Dev* 17: 615-628.
- Buisson R, Boisvert JL, Benes CH, Zou L. 2015. Distinct but Concerted Roles of ATR, DNA-PK, and Chk1 in Countering Replication Stress during S Phase. *Mol Cell* 59: 1011-1024.
- Crowe JL, Shao Z, Wang XS, Wei PC, Jiang W, Lee BJ, Estes VM, Alt FW, Zha S. 2018. Kinase-dependent structural role of DNA-PKcs during immunoglobulin class switch recombination. *Proc Natl Acad Sci U S A*.
- Crowe JL, Wang XS, Shao Z, Lee BJ, Estes VM, Zha S. 2020. DNA-PKcs phosphorylation at the T2609 cluster alters the repair pathway choice during immunoglobulin class switch recombination. *Proc Natl Acad Sci U S A* 117: 22953-22961.
- Forey R, Poveda A, Sharma S, Barthe A, Padioleau I, Renard C, Lambert R, Skrzypczak M, Ginalski K, Lengronne A et al. 2020. Mec1 Is Activated at the Onset of Normal S Phase by Low-dNTP Pools Impeding DNA Replication. *Mol Cell* 78: 396-410 e394.
- Franco S, Gostissa M, Zha S, Lombard DB, Murphy MM, Zarrin AA, Yan C, Tepsuporn S, Morales JC, Adams MM et al. 2006. H2AX prevents DNA breaks from progressing to chromosome breaks and translocations. *MolCell* 21: 201-214.
- Gonzalez Besteiro MA, Calzetta NL, Loureiro SM, Habif M, Betous R, Pillaire MJ, Maffia A, Sabbioneda S, Hoffmann JS, Gottifredi V. 2019. Chk1 loss creates replication barriers that compromise cell survival independently of excess origin firing. *EMBO J* 38: e101284.
- Gorthi A, Romero JC, Loranc E, Cao L, Lawrence LA, Goodale E, Iniguez AB, Bernard X, Masamsetti VP, Roston S et al. 2018. EWS-FLI1 increases transcription to cause R-loops and block BRCA1 repair in Ewing sarcoma. *Nature* 555: 387-391.
- Hart T, Tong AHY, Chan K, Van Leeuwen J, Seetharaman A, Aregger M, Chandrashekhara M, Hustedt N, Seth S, Noonan A et al. 2017. Evaluation and Design of Genome-Wide CRISPR/SpCas9 Knockout Screens. *G3 (Bethesda)* 7: 2719-2727.
- Hobeika E, Thiemann S, Storch B, Jumaa H, Nielsen PJ, Pelanda R, Reth M. 2006. Testing gene function early in the B cell lineage in mb1-cre mice. *Proc Natl Acad Sci U S A* 103: 13789-13794.
- Jiang W, Estes VM, Wang XS, Shao Z, Lee BJ, Lin X, Crowe JL, Zha S. 2019. Phosphorylation at S2053 in Murine (S2056 in Human) DNA-PKcs Is Dispensable for Lymphocyte Development and Class Switch Recombination. *J Immunol* 203: 178-187.

Klein U, Tu Y, Stolovitzky GA, Keller JL, Haddad J, Jr., Miljkovic V, Cattoretti G, Califano A, Dalla-Favera R. 2003. Gene expression dynamics during germinal center transit in B cells. *Ann N Y Acad Sci* 987: 166-172.

Lane AN, Fan TW. 2015. Regulation of mammalian nucleotide metabolism and biosynthesis. *Nucleic Acids Res* 43: 2466-2485.

Lee TH, Park JM, Leem SH, Kang TH. 2014a. Coordinated regulation of XPA stability by ATR and HERC2 during nucleotide excision repair. *Oncogene* 33: 19-25.

Lee Y, Brown EJ, Chang S, McKinnon PJ. 2014b. Pot1a prevents telomere dysfunction and ATM-dependent neuronal loss. *The Journal of neuroscience : the official journal of the Society for Neuroscience* 34: 7836-7844.

Liu X, Wang XS, Lee BJ, Wu-Baer FK, Lin X, Shao Z, Estes VM, Gautier J, Baer R, Zha S. 2019. CtIP is essential for early B cell proliferation and development in mice. *J Exp Med* 216: 1648-1663.

Lumsden JM, McCarty T, Petiniot LK, Shen R, Barlow C, Wynn TA, Morse HC, III, Gearhart PJ, Wynshaw-Boris A, Max EE et al. 2004. Immunoglobulin class switch recombination is impaired in Atm-deficient mice. *JExpMed* 200: 1111-1121.

Manis JP, Dudley D, Kaylor L, Alt FW. 2002. IgH class switch recombination to IgG1 in DNA-PKcs-deficient B cells. *Immunity* 16: 607-617.

McBride KM, Gazumyan A, Woo EM, Barreto VM, Robbiani DF, Chait BT, Nussenzweig MC. 2006. Regulation of hypermutation by activation-induced cytidine deaminase phosphorylation. *ProcNatlAcadSciUSA* 103: 8798-8803.

Menolfi D, Jiang W, Lee BJ, Moiseeva T, Shao Z, Estes V, Frattini MG, Bakkenist CJ, Zha S. 2018. Kinase-dead ATR differs from ATR loss by limiting the dynamic exchange of ATR and RPA. *Nature communications* 9: 5351.

Moiseeva TN, Qian C, Sugitani N, Osmanbeyoglu HU, Bakkenist CJ. 2019. WEE1 kinase inhibitor AZD1775 induces CDK1 kinase-dependent origin firing in unperturbed G1- and S-phase cells. *Proc Natl Acad Sci U S A* 116: 23891-23893.

Mokrani-Benhelli H, Gaillard L, Biasutto P, Le Guen T, Touzot F, Vasquez N, Komatsu J, Conseiller E, Picard C, Gluckman E et al. 2013. Primary microcephaly, impaired DNA replication, and genomic instability caused by compound heterozygous ATR mutations. *Hum Mutat* 34: 374-384.

Murga M, Bunting S, Montana MF, Soria R, Mulero F, Canamero M, Lee Y, McKinnon PJ, Nussenzweig A, Fernandez-Capetillo O. 2009. A mouse model of ATR-Seckel shows embryonic replicative stress and accelerated aging. *Nat Genet* 41: 891-898.

Noordermeer SM, Adam S, Setiাপutra D, Barazas M, Pettitt SJ, Ling AK, Olivieri M, Alvarez-Quilon A, Moatti N, Zimmermann M et al. 2018. The shieldin complex mediates 53BP1-dependent DNA repair. *Nature* 560: 117-121.

Olivieri M, Cho T, Alvarez-Quilon A, Li K, Schellenberg MJ, Zimmermann M, Hustedt N, Rossi SE, Adam S, Melo H et al. 2020. A Genetic Map of the Response to DNA Damage in Human Cells. *Cell* 182: 481-496 e421.

Onksen JL, Brown EJ, Blendy JA. 2011. Selective deletion of a cell cycle checkpoint kinase (ATR) reduces neurogenesis and alters responses in rodent models of behavioral affect. *Neuropsychopharmacology : official publication of the American College of Neuropsychopharmacology* 36: 960-969.

Pan-Hammarstrom Q, Lahdesmaki A, Zhao Y, Du L, Zhao Z, Wen S, Ruiz-Perez VL, Dunn-Walters DK, Goodship JA, Hammarstrom L. 2006. Disparate roles of ATR and ATM in immunoglobulin class switch recombination and somatic hypermutation. *JExpMed* 203: 99-110.

Polato F, Callen E, Wong N, Faryabi R, Bunting S, Chen HT, Kozak M, Kruhlak MJ, Reczek CR, Lee WH et al. 2014. CtIP-mediated resection is essential for viability and can operate independently of BRCA1. *J Exp Med* 211: 1027-1036.

Reina-San-Martin B, Chen HT, Nussenzweig A, Nussenzweig MC. 2004. ATM is required for efficient recombination between immunoglobulin switch regions. *JExpMed* 200: 1103-1110.

- Ruzankina Y, Pinzon-Guzman C, Asare A, Ong T, Pontano L, Cotsarelis G, Zediak VP, Velez M, Bhandoola A, Brown EJ. 2007. Deletion of the developmentally essential gene ATR in adult mice leads to age-related phenotypes and stem cell loss. *Cell Stem Cell* 1: 113-126.
- Ruzankina Y, Schoppy DW, Asare A, Clark CE, Vonderheide RH, Brown EJ. 2009. Tissue regenerative delays and synthetic lethality in adult mice after combined deletion of Atr and Trp53. *Nat Genet* 41: 1144-1149.
- Schrader CE, Linehan EK, Mochegova SN, Woodland RT, Stavnezer J. 2005. Inducible DNA breaks in Ig S regions are dependent on AID and UNG. *JExpMed* 202: 561-568.
- Shao Z, Flynn RA, Crowe JL, Zhu Y, Liang J, Jiang W, Aryan F, Aoude P, Bertozzi CR, Estes VM et al. 2020. DNA-PKcs has KU-dependent function in rRNA processing and haematopoiesis. *Nature* 579: 291-296.
- Sugitani N, Vendetti FP, Cipriano AJ, Pandya P, Deppas JJ, Moiseeva TN, Schamus-Haynes S, Wang Y, Palmer D, Osmanbeyoglu HU et al. 2022. Thymidine rescues ATR kinase inhibitor-induced deoxyuridine contamination in genomic DNA, cell death, and interferon-alpha/beta expression. *Cell reports* 40: 111371.
- Toy G, Austin WR, Liao HI, Cheng D, Singh A, Campbell DO, Ishikawa TO, Lehmann LW, Satyamurthy N, Phelps ME et al. 2010. Requirement for deoxycytidine kinase in T and B lymphocyte development. *Proc Natl Acad Sci U S A* 107: 5551-5556.
- Wang XS, Zhao J, Wu-Baer F, Shao Z, Lee BJ, Cupo OM, Rabadan R, Gautier J, Baer R, Zha S. 2020. CtIP-mediated DNA resection is dispensable for IgH class switch recombination by alternative end-joining. *Proc Natl Acad Sci U S A* 117: 25700-25711.
- Zimmermann M, Murina O, Reijns MAM, Agathangelou A, Challis R, Tarnauskaite Z, Muir M, Fluteau A, Aregger M, McEwan A et al. 2018. CRISPR screens identify genomic ribonucleotides as a source of PARP-trapping lesions. *Nature* 559: 285-289.

REVIEWER COMMENTS

Reviewer #1 (Remarks to the Author):

This revised version of the manuscript "ATR kinase supports normal proliferation in the early S phase by preventing replication resource exhaustion" by Menolfi et al. addresses some of the reviewers' comments and highlights some important controls. The level of novelty did not significantly change with the exception of the new data on the transformed B-cells. The changes are not highlighted in the text of the manuscript or quoted in the rebuttal, so it is difficult to assess how much it has changed.

During the revision, the authors generated v-abl kinase transformed B cell lines, capable of deleting ATR gene. In the rebuttal letter, the authors highlight the differences between the primary B cells and the transformed cells, implying that the non-transformed cells are more sensitive to ATR depletion. The phrasing in the revised article is much more restrained, talking about potentially more prominent role of ATR in primary cells. Assuming the goal of this experiment was to highlight the differences between the transformed and non-transformed cells, these differences should be discussed in detail, including the implications for the clinical use of ATR inhibitors – would they be more toxic to the immune system than to B-cell lymphomas? This is a novel result that clearly separates this study from the previous literature, and in my opinion, it deserved more attention and discussion, and potentially additional clarifying experiments – for example, with ATR inhibition.

In response to my comment #1 about various treatment times on figure 2f, the authors refer to a 1h treatment in the middle of S-phase on figure 3e that, as expected, had no effect on the ability of the cells to complete the cell cycle. However, my comment stated "ATRi treatment in early S should be limited to 2h", it was with regards to figure 2f, and this was not done. Figure 2f still compares 2h treatment to 48h treatment, and this is not very informative. If it is impossible to pinpoint the 2h of early S-phase, comparing the exposure for the first 24h only to the exposure for the second 24h (24-48) could be an option. A similar point was brought up by Reviewer #3, where the authors respond that "once the cells have started proliferation, ATR inhibition has less immediate impacts". The effects of ATR inhibition on proliferating S-phase cells have been studied repeatedly and it is clear that ATRi does not act like HU. What needs clarification is how "immediate" the effects on the early S-phase cells are. Because most of the treatments supporting this statement are 48h, which is not a short-term treatment. How do the authors discriminate between the effects of the exposure length vs the timing of the exposure to a particular cell cycle phase? I think this is a key point to support the conclusion about the role of ATR specifically in early S-phase.

In response to my comment #4, the authors added three relevant citations, although they picked a wrong citation from Bakkenist lab – the WEE1 inhibitor study instead of the ATR/CHK1 study. It is worth noting that two out of these three articles use siRNA-mediated CHK1 depletion rather than kinase inhibition, and the third one includes data on the non-cancer BJ-hTERT cells, contrary to what the authors state in the rebuttal ("the three references all use cancer cell lines treated with either small molecule inhibitors for ATR or CHK1") and in the main article text ("Indeed, several previous studies have tried to counteract the toxicity of ATR or CHK1 inhibitors by inhibiting replication origin firing with variable degrees of success in human cancer cell lines"). The three articles also focus on the effects of ATR/CHK1 depletion/inactivation on origin firing and fork speed, and hardly mention the toxicity of the inhibitors. Therefore, my comment about properly discussing relevant literature is not fully addressed.

In response to comment #6 by Reviewer #2, the authors claim there is no good antibody for mouse ATR, but there is definitely at least one antibody that has been repeatedly used to detect mouse ATR by western blot – cell signaling #2790. Multiple citations are listed on the manufacturer's website, including three published articles that successfully used it to detect mouse ATR by western blot.

Reviewer #2 (Remarks to the Author):

The authors have satisfyingly answered most my comments.

Reviewer #3 (Remarks to the Author):

The authors have addressed or attempted to address all the comments that were raised. This is now a really nice study ready for publication.

Point by Point response for the 2nd revision – Menolfi D. et al.

We thank the reviewers and the editor for their comments on our manuscript. We are delighted to find that Reviewer 2 and Reviewer 3 have now supported the publication of the revised manuscript. In this round of review, Reviewer 1 brought up additional comments, which we addressed in the revision with text clarifications, additional experiments, other references, and controls. We believe we have addressed all remaining remarks to the best of our ability. We agreed with reviewer 1 on the clinical significance of the tumor-normal difference, given the toxicities observed in ongoing clinical trials involving ATR inhibition. But we felt that such an important topic would require more in-depth study in better-controlled settings. We are reluctant to draw such a bold conclusion based on our limited cell line data. Please see specific comments for more details. Given the current manuscript's size and the data volume, we felt that such an important question would be better saved for future investigation. Our study represents the first study using a clean genetic deletion system to address the role of ATR in primary somatic cells. Our findings complement the other studies using ATR inhibitors or transient si/shRNA knockdowns in cell lines and primary cells. The extended G0 arrest in primary B cells ensured a complete deletion. The rapid response to cytokine-induced cell cycle re-entry provides a unique system to study the role of ATR, an essential gene in proliferating cells, in undisturbed replication. The comprehensive metabolism and RNA-seq analyses at the defined G0, G1, and first S phases systematically characterized ATR's impact in early B cell activation, having broad implications for lymphocyte activation and somatic cell proliferation. Moreover, the similarity between the ATR null and ATR kinase dead models assigned this function of ATR to its kinase activity. As such, we think the information would be valuable for DNA replication and stress, genome stability, checkpoint kinases, metabolism, and immune activation. Comments from the reviewers are *italicized* below for easy identification.

REVIEWER COMMENTS

Reviewer #1 (Remarks to the Author):

This revised version of the manuscript "ATR kinase supports normal proliferation in the early S phase by preventing replication resource exhaustion" by Menolfi et al. addresses some of the reviewers' comments and highlights some important controls. The level of novelty did not significantly change with the exception of the new data on the transformed B-cells. The changes are not highlighted in the text of the manuscript or quoted in the rebuttal, so it is difficult to assess how much it has changed.

In this 2nd revision, we have included a text file **highlighting** (yellow) the major text changes.

During the revision, the authors generated v-abl kinase transformed B cell lines, capable of deleting ATR gene. In the rebuttal letter, the authors highlight the differences between the primary B cells and the transformed cells, implying that the non-transformed cells are more sensitive to ATR depletion. The phrasing in the revised article is much more restrained, talking about potentially more prominent role of ATR in primary cells. Assuming the goal of this experiment was to highlight the differences between the transformed and non-transformed cells, these differences should be discussed in detail, including the implications for the clinical use of ATR inhibitors – would they be more toxic to the immune system than to B-cell lymphomas? This is a novel result that clearly separates this study from the previous literature, and in my opinion, it deserved more attention and discussion, and potentially additional clarifying experiments – for example, with ATR inhibition.

We are encouraged by Reviewer 1's enthusiasm for the difference between v-abl transformed B cells vs. primary B cells in our supplementary data. We agree that the tumor vs. normal difference would be valuable for developing and using ATR kinase inhibitors in cancer therapy. For this reason, we also felt that it is a topic that deserves a separate, better controlled, and much more in-depth study. We were asked to compare the tumor vs. normal differences on the primary review and performed this set of experiments. While the results are potentially interesting, they are not sufficient to support a general statement on cancer vs. normal difference, given only one type of transformed B cells (v-abl cells) has been tested. There are many differences between the v-abl kinase transformed pro-/pre-B cells vs. the primary naïve IgM+ splenic B cells beyond their transformation status that might contribute to this difference. These include but are not limited to 1) the developmental stage (pre/proB vs. mature IgM+ B cells), 2) the requirement for B cell receptor signaling

(not required in the v-abl cells, highly dependent on the naïve B cells), 3) the cell cycle status that precedes the DNA replication (G0 for naïve B cells, and G1 at most for continue cultured v-abl cells), and 4) other genetic differences, including frequent inactivation of the p53 pathway during v-abl kinase transformation documented in prior studies (1, 2). For these reasons, we only speculate briefly in one sentence and leave it for future study. We hope the reviewer will understand our cautions here.

In response to my comment #1 about various treatment times on figure 2f, the authors refer to a 1h treatment in the middle of S-phase on figure 3e that, as expected, had no effect on the ability of the cells to complete the cell cycle. However, my comment stated "ATRi treatment in early S should be limited to 2h", it was with regards to figure 2f, and this was not done. Figure 2f still compares 2h treatment to 48h treatment, and this is not very informative. If it is impossible to pinpoint the 2h of early S-phase, comparing the exposure for the first 24h only to the exposure for the second 24h (24-48) could be an option. A similar point was brought up by Reviewer #3, where the authors respond that "once the cells have started proliferation, ATR inhibition has less immediate impacts". The effects of ATR inhibition on proliferating S-phase cells have been studied repeatedly and it is clear that ATRi does not act like HU. What needs clarification is how "immediate" the effects on the early S-phase cells are. Because most of the treatments supporting this statement are 48h, which is not a short-term treatment. How do the authors discriminate between the effects of the exposure length vs the timing of the exposure to a particular cell cycle phase? I think this is a key point to support the conclusion about the role of ATR specifically in early S-phase.

We apologize for not clarifying this feasibility challenge to the reviewer on the first revision. As shown in the Sup Fig. S3a (copied below), the primary cells enter the S phase GRADUALLY during a ~12-14 hours window (from 12 to 24hr after stimulation). The *Atr*-deficient cells often initiate DNA replication/enter the S phase slightly slower than the *Atr*-proficient counterparts. Therefore, it is NOT feasible to consistently identify "early S phase cells within any given two-hour window" in this system. The early S phase for one subset of the cells in one experiment would include substantial mid-S or pre-S phase cells for another experiment. We cannot distinguish the early vs. mid-phase cells even if we use BrdU "label and chase" to exclude the pre-replicating (before S phase) cells. We hope this explanation is now clear.

Sup Figure S3a

As for the alternative approach proposed by the reviewer, in Figure 3e (copied below), we compare the first 24hr (0-24hr) treatment vs. the 2nd 24hr (24-48hr ATR inhibitor) treatment. Indeed, loss of *Atr* kinase activity in the first 24hr (either by the deletion in the *Atr*^{C/-} cells or by inhibition via ATRi from time zero -blue line) has a much more profound impact on DNA replication upon pulse-chase - 14% of BrdU+ cells returned to G1 in 6 hr in the 24-48hr treatment group, vs. 0.87% for genetic deletion and 1.87% for ATRi for 0-24hr. Based on these observations, we decided to focus the rest of the analyses on the first 24hr. We reason that the low levels of replication factors in the G0 cells pose a significant challenge to the ATR-deficient cells in their first cell division when too many origins are fired simultaneously. In the subsequent cell cycles (*e.g.*, 24-48hr), the cells have significant leftover nucleotides and other replication factors from the last S phase, thus less sensitive to origin over-firing associated with ATR-kinase deficiency.

In response to my comment #4, the authors added three relevant citations, although they picked a wrong citation from Bakkenist lab – the WEE1 inhibitor study instead of the ATR/CHK1 study. It is worth noting that two out of these three articles use siRNA-mediated CHK1 depletion rather than kinase inhibition, and the third one includes data on the non-cancer BJ-hTERT cells, contrary to what the authors state in the rebuttal ("the three references all use cancer cell lines treated with either small molecule inhibitors for ATR or CHK1") and in the main article text ("Indeed, several previous studies have tried to counteract the toxicity of ATR or CHK1 inhibitors by inhibiting replication origin firing with variable degrees of success in human cancer cell lines"). The three articles also focus on the effects of ATR/CHK1 depletion/inactivation on origin firing and fork speed, and hardly mention the toxicity of the inhibitors. Therefore, my comment about properly discussing relevant literature is not fully addressed.

We apologize for the wrong citation. The citation has been corrected (ref 62 in the main text). These studies in human immortalized cell lines or cancer cells in continuous culture documented an important role of ATR and CHK1 kinases in control origin firing and replication initiation by counteracting CDC7 or CDK1 activities (3-5) (ref, 38, 61 and 62 in the main text). As pointed out by the reviewer, these studies did not identify origin firing error as the cause of ATR inhibition toxicity, a primary focus of our research. We have now modified the text to:

"Indeed, several previous studies in human immortalized or cancer cells have suggested a function of ATR and CHK1 in controlling replication initiation and origin firing by counteracting CDC7 or CDK1 activities (3-5). Given that factors implicated in original firing frequently appear in our CRISPR survival screen, we set out to test whether suppressing origin firing could restore productive replication in primary cells with clean deletion of ATR."

In response to comment #6 by Reviewer #2, the authors claim there is no good antibody for mouse ATR, but there is definitely at least one antibody that has been repeatedly used to detect mouse ATR by western blot – cell signaling #2790. Multiple citations are listed on the manufacturer's website, including three published articles that successfully used it to detect mouse ATR by western blot.

We thank the reviewer for identifying this antibody. Indeed, this was precisely the antibody (Cell signaling #2790) that we had used in our 2018 publication (Nature Communication, Menolfi, et al. 2018)(6) to detect mouse ATR with cell lines

carrying identical Atr conditional alleles. The relevant SupFig.1C from this 2018 paper is on the left. This antibody is a polyclonal rabbit antibody. The lot used in our 2018 paper (purchased in 2017) was No. 7. The one after 2019 is No.10. The newer one failed to detect mouse ATR in multiple attempts. Consistent with our experience, out of the 160 references the Cell Signaling website listed for this antibody, only 5 were for mice. All 5 (copied on the right) were submitted for publication on or before 2019. The

Elife ATR expands embryonic stem cell fate potential in response to replication stress. Sina Atashpaz, et. al. 2020	Applications:Western Blotting (WB) Reactivity:Mus musculus (House mouse)
Eneuro The Progesterone Receptor Interactome in the Female Mouse Hypothalamus: Interactions with Synaptic Proteins Are Isoform Spe... Kalpana D Acharya, et. al. 2017	Applications:Unspecified Reactivity:Mus musculus (House mouse)
Autophagy ABHD5 interacts with BECN1 to regulate autophagy and tumorigenesis of colon cancer independent of PNPLA2. Yuan Peng, et. al. 2016	Applications:Unspecified Reactivity:Mus musculus (House mouse)
Cell Cycle Wild-type p53-induced phosphatase 1 (Wip1) forestalls cellular premature senescence at physiological oxygen levels by re... Hiroyasu Sakai, et. al. 2014	Applications:Western Blotting (WB) Reactivity:Mus musculus (House mouse)
Cell Death Differ E2f2 induces cone photoreceptor apoptosis independent of E2f1 and E2f3. D Chen, et. al. 2013	Applications:Western Blotting (WB) Reactivity:Mus musculus (House mouse)

latest one was submitted in 2019 and published in 2020 (<https://elifesciences.org/articles/54756>). All others were published before 2017.

Nevertheless, we now included a Western blot with the leftover Lot 7 antibody in the lab on purified CD43- splenocytes (~90% B cells) and control thymocytes (thymus) from the same mice. The results supported a clear reduction of ATR protein in the B cells but not T cells from the *CD21Cre+Atr^{C/-}* mice (compare lanes 4 and 6).

Moreover, with human 293T cells and RPE1 cells as a control, the result suggests that the expression level of ATR protein might be much lower in mouse cells than in human cells. Similar observations have been made for DNA-PKcs, a related PI3 Kinase-like protein kinase family member. We now included the data in Figure S1b (on the right here).

In addition to protein expression, we had also taken multiple measurements to verify the clean deletion of ATR in the naïve B cells characterized here. 1st, this is a well-characterized ATR conditional allele that was made and published in 2003 by Dr. Eric Brown and colleagues (7-9). 2nd, we have published this ATR conditional allele with clean Western blotting for deletion in Nature Communications (2018 Menolfi D et. al) (6). 3rd, in this manuscript, we included sensitive PCR assays for clean deletion at the DNA level (Fig. S1), and 4th, we described an extreme phenotype for the Mb1Cre mediated deletion of ATR in immature B cells and also a lethal phenotype of CD21Cre mediated ATR deletion in activated primary cells - dying within 24hr of activation. We are confident that ATR is truly deleted for all the above reasons.

Reviewer #2 (Remarks to the Author):

The authors have satisfyingly answered most my comments.

Reviewer #3 (Remarks to the Author):

The authors have addressed or attempted to address all the comments that were raised. This is now a really nice study ready for publication.

We thank Reviewer 2 and Reviewer 3 for their suggestions and support that helped us improve our manuscript.

Reference

1. F. Cong, X. Zou, K. Hinrichs, K. Calame, S. P. Goff, Inhibition of v-Abl transformation by p53 and p19ARF. *Oncogene* 18, 7731-7739 (1999).
2. I. Unnikrishnan, A. Radfar, J. Jenab-Wolcott, N. Rosenberg, p53 mediates apoptotic crisis in primary Abelson virus-transformed pre-B cells. *Mol Cell Biol* 19, 4825-4831 (1999).
3. M. A. Gonzalez Besteiro *et al.*, Chk1 loss creates replication barriers that compromise cell survival independently of excess origin firing. *EMBO J* 38, e101284 (2019).
4. E. Petermann, M. Woodcock, T. Helleday, Chk1 promotes replication fork progression by controlling replication initiation. *Proc Natl Acad Sci U S A* 107, 16090-16095 (2010).
5. T. N. Moiseeva *et al.*, An ATR and CHK1 kinase signaling mechanism that limits origin firing during unperturbed DNA replication. *Proc Natl Acad Sci U S A* 116, 13374-13383 (2019).
6. D. Menolfi *et al.*, Kinase-dead ATR differs from ATR loss by limiting the dynamic exchange of ATR and RPA. *Nature communications* 9, 5351 (2018).
7. E. J. Brown, D. Baltimore, Essential and dispensable roles of ATR in cell cycle arrest and genome maintenance. *Genes Dev* 17, 615-628 (2003).
8. Y. Ruzankina *et al.*, Tissue regenerative delays and synthetic lethality in adult mice after combined deletion of Atr and Trp53. *Nat Genet* 41, 1144-1149 (2009).
9. H. Rojo *et al.*, ATR acts stage specifically to regulate multiple aspects of mammalian meiotic silencing. *Genes Dev* 27, 1484-1494 (2013).

REVIEWERS' COMMENTS

Reviewer #1 (Remarks to the Author):

In this revised version the authors addressed some of the shortcomings pointed out by the previous review rounds. I believe the article can be published, provided the following problems are fixed:

1. Figure 2f should be removed. Comparing 2h incubation with 48h incubation is not informative at all, and since the authors explained that finding the right 2h is impossible, this experiment has no value and only adds confusion.

2. The authors claim that they compared 0-24 h and 24-48 h incubations, however the labelling is not in agreement with this statement. The schematic in the upper left corner clearly states that the blue ATRi panels are treated for 48 h: 0 to 48h, but the orange arrow next to it in the rebuttal says "0-24h". Which one is it? As far as I can see, it is 0-48h, which has been there since the first submission, and the figure legend is in agreement with this ("cells were treated with 5 μ M ATRi (VE-821) supplemented at the beginning of the experiment (blue treatment), or at 24 h post-stimulation (red treatment) until collection"). But then there's actually no 0-24h incubation in this experiment, so there's still no comparison of treatments of equal lengths. "14% of BrdU+ cells returned to G1 in 6 hr in the 24-48hr" means these cells have only been treated with ATR for 6 hours at the time of harvest, and the ones in G1 probably spent most of this time outside of S-phase, so the effect on S-phase cannot be deduced from this, and certainly the authors can't compare this sample to the cells that have been treated with ATRi for 30h before the harvest.

I understand, that this is not an easy problem to solve, but at the very minimum it has to be clearly and explicitly discussed in the manuscript as a limitation. The effect on the early S cells is one of the key conclusions of this article, and the best experiment supporting it is based on comparing 24h treatment to 48h treatment, which are not equal. Alternative explanations for the observed phenotypes should be discussed.

We thank reviewer 1 for his/her patience and carefulness. In the response, we provide our reasons for why Figure 2f is important for the integrity of the manuscript and made a critical point about ATR being essential for productive DNA replication ONLY IF ATR is inhibited or lost before S phase, not during S phase. In the 3rd revision of this manuscript, we have now better explained and labeled the BrdU chase experiment. At the time of BrdU labeling, the cells have experienced 0-24hr of ATR inhibition. This is the population that we have "chased after". During the chase phase, ATR inhibitors are still present in the culture. In the other two experiments, the cells have only experienced 30 minutes of ATRi at the time of BrdU labeling and were chased with or without ATRi for 24hr. Details can be found below. All the original comments from the editors and reviewer 1 are italicized for easy identification.

Editor's points

We therefore invite you to revise your paper one last time to address the remaining concerns of our reviewers and our editorial requests in the attached document(s). Please explain here the rationale for the chosen time points and comparisons and clearly state limitations of Fig 2f. Please make it clear for whether you compare 0-24h or 24-48h in Fig. 2, Fig.3 and SFig.3a.

We thank the editor for giving us this chance to clarify ourselves one more time. We have included all the editorial modifications/files requested in the revision. In response to reviewer 1 below, we clearly stated the value of Fig. 2f. The experiments made the important point, which we also discuss later, that inhibition of ATR in "S phase" cells (already BrdU+) does NOT abrogate replication. Therefore, ATR has to be inhibited before full bloom S phase starts. We had made it clear in the figure legend about how long we treated the cells with ATRi and when they were treated and when the BrdU labeling occurred (in the pulse chase experiments).

REVIEWERS' COMMENTS

Reviewer #1 (Remarks to the Author):

In this revised version the authors addressed some of the shortcomings pointed out by the previous review rounds. I believe the article can be published, provided the following problems are fixed:

1. Figure 2f should be removed. Comparing 2h incubation with 48 incubation is not informative at all, and since the authors explained that finding the right 2h is impossible, this experiment has no value and only adds confusion.

We thank reviewer 1 for her positive assessment of our 2nd revised manuscript and for his/her patience.

Maybe we were not clear in our manuscript writing. Figure 2f is critical for the integrity of the manuscript and the main message of the study, for at least three reasons.

First, this two-hour ATR inhibitor incubation and the lack of impact on replication measured by BrdU shows that ATR and its kinase activity are NOT required for on-going replication. This is very different from the role of ATR in replication stress response, where ATR inhibition immediately before the damage (e.g., HU) would abolish the subsequent replication stress response.

Second, it is precisely this result that led us to investigate when ATR must be inhibited to jeopardize replication, therefore move the timepoints to the first 24hr (in the following experiments) and eventually identify a critical role of ATR in the initiation of the first S phase.

Third, this Figure panel serves as an important control for the specificity of ATR in physiological DNA replication. As such treating the cells with ATM inhibitor, DNA-PKcs inhibitor or HU (showed in Figure S2g) did not create the same effect.

For all these reasons, we believe that Figure 2f has earned its space in this manuscript and in the main figure panel.

2. The authors claim that they compared 0-24 h and 24-48 h incubations, however the labelling is not in agreement with this statement. The schematic in the upper left corner clearly states that the blue ATRi panels are treated for 48 h: 0 to 48h, but the orange arrow next to it in the rebuttal says “0-24h”. Which one is it? As far as I can see, it is 0-48h, which has been there since the first submission, and the figure legend is in agreement with this (“cells were treated with 5 μ M ATRi (VE-821) supplemented at the beginning of the experiment (blue treatment), or at 24 h post-stimulation (red treatment) until collection”). But then there’s actually no 0-24h incubation in this experiment, so there’s still no comparison of treatments of equal lengths. “14% of BrdU+ cells returned to G1 in 6 hr in the 24-48hr” means these cells have only been treated with ATR for 6 hours at the time of harvest, and the ones in G1 probably spent most of this time outside of S-phase, so the effect on S-phase cannot be deduced from this, and certainly the authors can’t compare this sample to the cells that have been treated with ATRi for 30h before the harvest.

I understand, that this is not an easy problem to solve, but at the very minimum it has to be clearly and explicitly discussed in the manuscript as a limitation. The effect on the early S cells is one of the key conclusions of this article, and the best experiment supporting it is based on comparing 24h treatment to 48h treatment, which are not equal. Alternative explanations for the observed phenotypes should be discussed.

We apologize for not making ourselves clear in the prior response to the review. Since those experiments in Figure 3e are “pulse”-chase experiments, in which we follow the S phase cells that were pulse labeled with BrdU. For the first row – blue line in the diagram, at the time of labeling, the BrdU+ cells have been exposed to ATR inhibitor for only 24hr (therefore 0-24h). They were continually exposed to ATRi during the 24-48hr phase period. In the 2nd (red line in the diagram) and the 3rd row (green line in the diagram), the BrdU+ (labeled) cells were only exposed to ATRi for ~30mins at the time of labeling. They were either chased for another 24hr (from 24-48hr) in the presence of ATR inhibitor (red line) or without ATR inhibitor (green line in the diagram). In either case, ATR inhibition has very little impact on the continued cycling of those (already S phase) BrdU+ cells (no expose to ATRi in pre-S or G1 phase).

We have now modified the diagram to highlight the BrdU Pulse duration with larger fonts and open arrowhead. We also expanded the label to the right here for clarity. We also modified the figure legend to highlight the time of pulse chase and the ATR inhibitor exposure at the time of BrdU labeling. This does not change the interpretation of the data. Exposure to ATR inhibition or loss during S phase does NOT abrogate replication. And experiencing ATR inhibition/loss in the proceeding G1 is necessary for replication defects.